# Elements and evolutionary determinants of genomic divergence between paired primary and metastatic tumors

**Ruping Sun** [1,2☯] *, **Athanasios N. Nikolakopoulos** [1,2☯]

**1** Department of Laboratory Medicine and Pathology, University of Minnesota, Minneapolis, Minnesota, United States of America, **2** Masonic Cancer Center, University of Minnesota, Minneapolis, Minnesota, United States of America

☯ These authors contributed equally to this work.
* ruping@umn.edu

**Data Availability Statement:** All relevant data are within the manuscript and its Supporting information files.

## Abstract

Can metastatic-primary (M-P) genomic divergence measured from next generation sequencing reveal the natural history of metastatic dissemination? This remains an open question of utmost importance in facilitating a deeper understanding of metastatic progression, and thereby, improving its prevention. Here, we utilize mathematical and computational modeling to tackle this question as well as to provide a framework that illuminates the fundamental elements and evolutionary determinants of M-P divergence. Our framework facilitates the integration of sequencing detectability of somatic variants, and hence, paves the way towards bridging the measurable between-tumor heterogeneity with analytical modeling and interpretability. We show that the number of somatic variants of the metastatic seeding cell that are experimentally undetectable in the primary tumor, can be characterized as the path of the phylogenetic tree from the last appearing variant of the seeding cell back to the most recent detectable variant. We find that the expected length of this path is principally determined by the decay in detectability of the variants along the seeding cell's lineage; and thus, exhibits a significant dependence on the underlying tumor growth dynamics. A striking implication of this fact, is that dissemination from an advanced detectable subclone of the primary tumor can lead to an abrupt drop in the expected measurable M-P divergence, thereby breaking the previously assumed monotonic relation between seeding time and M-P divergence. This is emphatically verified by our single cell-based spatial tumor growth simulation, where we find that M-P divergence exhibits a non-monotonic relationship with seeding time when the primary tumor grows under branched and linear evolution. On the other hand, a monotonic relationship holds when we condition on the dynamics of progressive diversification, or by restricting the seeding cells to always originate from undetectable subclones. Our results highlight the fact that a precise understanding of tumor growth dynamics is the *sine qua non* for exploiting M-P divergence to reconstruct the chronology of metastatic dissemination. The quantitative models presented here enable further careful evaluation of M-P divergence in association with crucial evolutionary and sequencing parameters.

**Funding:** This work was supported by the Department of Laboratory Medicine and Pathology (https://med.umn.edu/pathology) and the Masonic Cancer Center (https://cancer.umn.edu/) at the University of Minnesota. This work was also supported by the Karen Wyckoff Rein in Sarcoma Foundation (https://www.reininsarcoma.org/). The funders had no role in study design, data collection and analysis, decision to publish, or preparation of the manuscript.

**Competing interests:** The authors have declared that no competing interests exist.

## Author summary

To properly translate the genomic divergence between paired metastatic and primary tumors (M-P divergence) into the natural history of metastatic spread, it is fundamental to understand what exactly is being captured on the trees of tumor evolution by such divergence. Here, we take the first step towards analytically decomposing the elements of the M-P divergence, and reveal the underlying principles that govern the relation between M-P divergence, clonal dynamics, and detectability of somatic variants in the primary tumor. In parallel to our mathematical framework, we further investigate the patterns of M-P divergence by performing single-cell-based spatial tumor growth simulation studies. Both methods demonstrate that the growth mode of the primary tumor modulates the dependence of M-P divergence on the metastatic dissemination time, which has important implications for the accurate translation of the measured divergence. Our study paves the way towards bridging the measurable between-tumor heterogeneity with analytical modeling and interpretability.

## 1 Introduction

Reconstructing the chronology of metastatic spread in human cancers [1–4] has profound clinical importance [5, 6] and significant potential in improving cancer treatment. It is an indispensable first step towards a deeper understanding of metastatic tumor progression. Evaluation of the genomic divergence between paired metastatic and primary tumors (M-P divergence) has become the *de facto* starting point for tackling this challenging task. However, despite the imperative need for accurate translation of M-P divergence measurements to metastatic seeding times, disparate interpretations of such measurements still abide. Should we read a small M-P divergence as an indication of "early" metastatic seeding [1, 7], or rather, as evidence that metastatic potential is acquired at a late stage [4, 8, 9] of primary tumor growth? This remains an open and challenging question. Before unraveling this puzzle, however, one needs to carefully characterize *what exactly is being captured* on the trees of tumor evolution by M-P divergence measurements arising from next generation sequencing.

Somatic evolution takes years to decades before the founder cell of the primary tumor gets established [10]. As it is a descendant of the founder that will be equipped with the ability to leave the primary site through the initial steps of the invasion-metastasis cascade [11], the dissemination event must happen late, in absolute terms. However, with the establishment of the founder granted, the connection between the mode of primary tumor expansion [12, 13] and the relative timing of metastatic seeding remains unclear. Moreover, the distinction between the initiation of the founder and subsequent expansion is also blurred; for example, the definition of the tumor founder can be fuzzy, given the fact that it is still challenging to identify the true "clonal" variants using a limited number of tumor samples [14, 15].

While the expansion dynamics of the primary tumor *in vivo* are largely undetectable, several hypothetical modes of growth have been proposed. Namely, effective neutrality mode, where clonal composition progressively diversifies [16]; branching evolution mode, where several subclones appear late and dominate the growth [17]; as well as, linear evolution mode with continuous clonal replacement [12]. The latter two modes differ from progressive diversification due to the presence of ongoing clonal sweeps at regional (branching evolution) or broad (linear evolution) scales, which leads to patterns of sub-clonal convergence rather than divergence. These modes represent possible tumor growth dynamics, which are determined by evolutionary and ecological forces, such as selection upon (epi)genetic alterations [18, 19],

spatial structure [13, 20], as well as, tumor micro-environments [21–23]. Accordingly, different tumor types and even different stages of a tumor may involve various growth dynamics [24–26].

To correctly translate experimental measurements of M-P divergence into cancer cell's migration history, it is necessary to decompose its intrinsic elements and understand their relation to the metastatic dissemination time under different tumor growth dynamics. A recently published deme (or compartment) based spatial tumor growth model suggests that small divergence reflects early metastatic seeding [1]. A caveat, however, is that deme-based models (by treating 5,000-10,000 genetically homogeneous cells as a single unit) are tailored to colorectal cancers and can translate only approximately the fitness of single cells (individuals) into deme-based growth rates [27]; furthermore the currently deployed strategy of randomly picking one compartment at a time for division [28] imposes additional limitations on the expressiveness and the biological plausibility of the model. By exploring limited conditions, the deme-based model mainly depicts the patterns of progressive diversification [13, 18]. Hence, other growth dynamics, such as branching and linear evolution, have not been investigated for this question. More importantly, despite some qualitative discussions [1, 9], a careful mathematical analysis relating measurable M-P divergence to the underlying phylogenetic tree patterns under different tumor growth dynamics, is still missing in the literature.

Genomic heterogeneity among tumor cells is notably fueled by the passenger somatic variants, which occur several orders of magnitude more frequently than the driver variants during cell divisions [29]. On the other hand, the bulk sequencing of tumor samples can only reveal variants that are present in a detectable fraction of cancer cells. Therefore, in conjunction with the underlying evolutionary forces, the observed M-P divergence is actively shaped by the experimental limitations in detectability of somatic variants as well. Multi-region sampling of longitudinal tumor samples emerges as a powerful tool to resolve the spatial and temporal heterogeneity patterns during cancer evolution [30–32]. However, the convolution between the detection process itself and the underlying evolutionary process, could potentially confound the conclusions drawn by solely analyzing the multi-region sequencing data. This is evidenced by the disparate interpretations of M-P divergence in relation to the metastatic seeding time. Overcoming such challenge calls for additional theoretical studies that aim to decompose the genomic divergence measures into their fundamental components. Our motivation, in this work, is aligned with explorations of the utility of phylogenetic metrics in informing particular evolutionary parameters [33]; however, we also aim to illuminate the intrinsic complexities of the system under study, by highlighting the evolutionary determinants of M-P divergence (Fig 1), as well as, the circumstances under which the divergence may be informative for inferring relevant parameters. As such, we take the first step towards a quantitative characterization of measurable M-P divergence as a function of seeding time and sequencing detectability; our analysis starts from the establishment of the founder, and our approach involves both mathematical modeling of the fundamental M-P divergence patterns under distinct tumor growth dynamics, as well as computational modeling of spatial growth kinetics subject to elaborate local density constraints (Fig 1).

## 2 Results

### 2.1 The elements of M-P divergence on the clonal evolutionary tree

We assume that the primary tumor growth begins with a single founder cell [34]. We focus on monoclonal seeding [35] at a time during the growth of the primary, i.e., individual progeny cells of the founder disseminate when the primary tumor reaches a size of $N_d$ (see Methods). The terms "dissemination" and "seeding" are used interchangeably in this paper as we do not

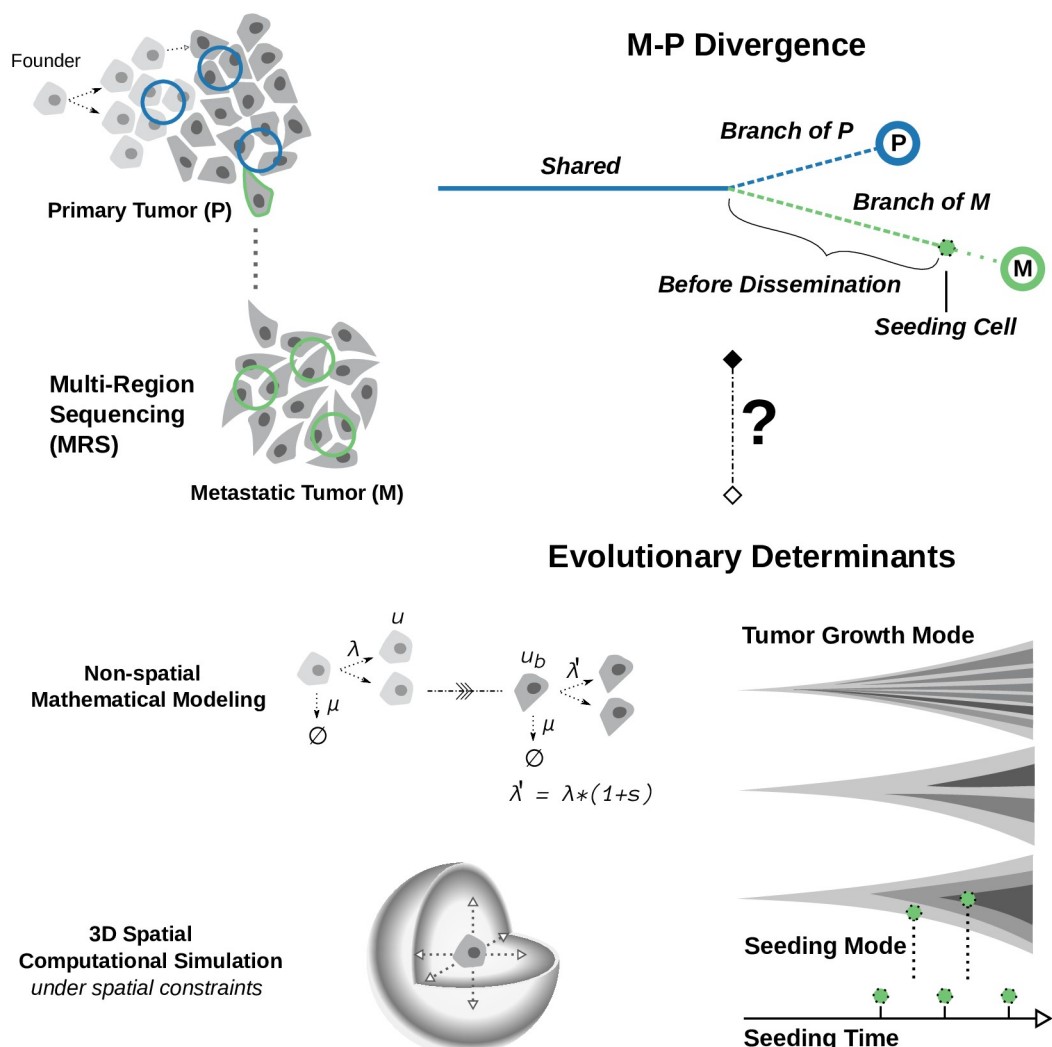

**Fig 1. The main research question and the study design.** Quantifying and interpreting the genomic divergence between paired primary and metastatic tumors (M-P divergence, top right panel) utilizing the ever-growing amount of multi-region/longitudinal cancer sequencing data (top left panel) is becoming increasingly important. It can facilitate a deeper understanding of metastatic progression, and thereby, improve its prevention. Here, we sought to ask what exactly is being captured on the trees of tumor evolution by M-P divergence measured in such datasets. We adopt mathematical modeling (non-spatial multi-type branching processes) to illuminate the key evolutionary determinants of M-P divergence; and computational modeling to simulate the three dimensional cancer clonal evolution, and we investigate how the M-P divergence is regulated by the evolutionary forces under various growth dynamics of the primary tumor (bottom panel). The detailed model setting, including the parameters used in both modeling approaches, can be found in Methods and Table 1.

model the processes in between. For models regarding other parts of the metastatic cascade, see [36, 37]. The primary and the single-clone-seeded metastatic tumor are detected until they reach size $N$, respectively. On the phylogenetic tree built from metastatic and primary tumor sample(s), the M-P divergence typically involves the metastatic specific Branch ($B$), i.e., number of (#) somatic variants ($v$, referred to as variants hereafter) found in the metastasis ($m$) but not in the primary ($p$): $B_m = \#\{v|f_v^m > \gamma, f_v^p < \alpha\}$ and the primary specific branch $B_p = \#\{v|f_v^p > \gamma, f_v^m < \alpha\}$, respectively [1, 9, 38] (Fig 2A). Here $\gamma$ is the variant allele frequency threshold above which substantial presence of a variant may be deemed. We assume that the

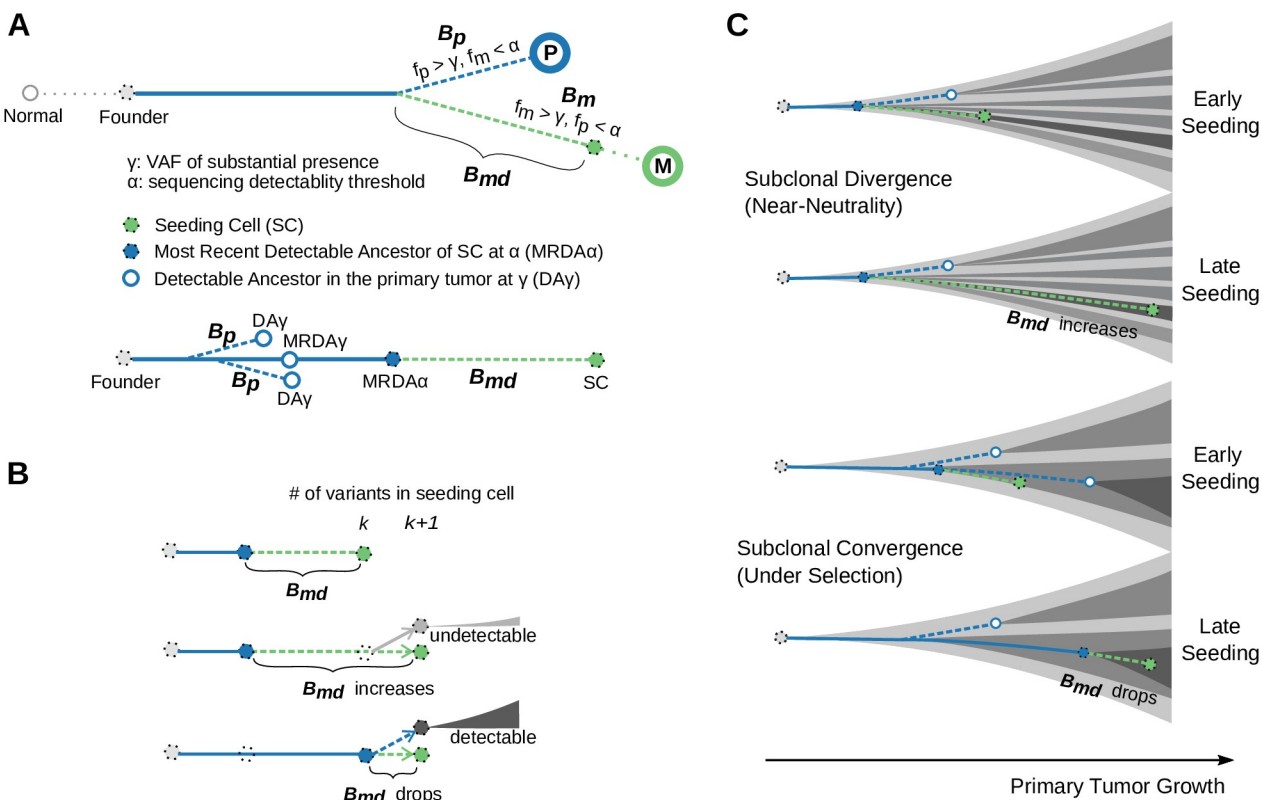

**Fig 2. Elements of Metastatic-Primary genomic divergence and their potential pattern in tumor evolution.** (**A**) A sample tree (schematic) based on somatic variant analysis of paired Metastatic (M) and Primary (P) tumor samples depicts the two branches involving the number of metastatic specific ($B_m$, dashed green line) and primary specific somatic variants ($B_p$, dashed blue line), respectively. We focus primarily on the variants accumulated in the metastatic seeding cell (green circle) before its dissemination ($B_{md}$), as it is directly related to the primary tumor growth and the genetic determinants of metastatic dissemination. Below the sample tree shown a genealogy tree focusing on the lineage of the seeding cell, where we note the branches that represent $B_{md}$ and $B_p$, respectively. Detectable ancestors in the primary at a frequency of $\gamma$ ($DA\gamma$) and the most recent detectable ancestor at $\alpha$ and $\gamma$ (MRDA$\alpha$ and MRDA$\gamma$) of the seeding cell are also indicated, respectively. Here $\gamma$ is the variant allele frequency threshold above which substantial presence of a variant may be deemed; $\alpha$ is the frequency threshold of sequencing detectability. (**B**) If a cell with $k$ variants "cancels" its dissemination (i.e., stays in the primary tumor) and its progeny cell with $k + 1$ variants migrates instead, the resulting $B_{md}$ can drop if the progenitor cell with $k$ variants eventually grows into a detectable subclone in the primary tumor. In this case, it becomes the most recent detectable ancestor of the seeding cell. More generally, $B_{md}$ can decrease if an ancestor along the genealogy the the seeding cell becomes detectable due to subclone expansion. (**C**) We hypothesize that $B_{md}$ and $B_p$ are affected by the kinetics of primary tumor growth. Under the growth pattern of progressive subclonal divergence, late seeding would lead to larger $B_{md}$ than early seeding as only the early subclones are detectable; $B_p$ stays constant as the detectable portion of the seeding cell lineage is restricted. By contrast, detectable subclones can appear late leading to the pattern of sub-clonal convergence (due to selection or other ecological forces), under which scenario seeding from early or late detectable subclones may lead to similar $B_{md}$, but differs in $B_p$.

variant detection is fair, i.e., any variant in the tumor with a frequency above a detectability threshold $\alpha$ will be detected e.g., through extensive multi-region sequencing [32]. Note that $\gamma$ and $\alpha$ are introduced to deal with the noise of bulk sequencing data in practice. For a variant to be considered condition-specific (i.e., primary- or metastatic-specific), one would avoid overestimating the divergence (e.g., control for false positive rates) by requiring that (1) the variant allele frequency is higher than a threshold $\gamma$ suggesting the substantial presence in a condition; and (2) the frequency in the other condition is lower than a threshold $\alpha$ suggesting the absence. Also note that $\gamma$ is greater than $\alpha$. The variants are assumed to be heterozygous, i.e., the cancer cell fraction thresholds will be at $2\gamma$ and $2\alpha$, respectively. While the somatic variants usually refer to point mutations and small indels, the framework could be extended for large structural changes and copy number alterations as well. $B_m$ can be further divided into

variants accumulated before ($B_{md}$) or after dissemination. We focus on $B_{md}$ as it is related to the primary tumor growth and the genetic determinants of dissemination [5]. $B_{md}$ refers to those variants that are NOT detectable in the primary tumor. $B_p$, on the other hand, includes all the variants with a frequency above $\gamma$ in the primary tumor except those appearing on the metastatic seeding cell. To illustrate $B_{md}$ and $B_p$ on the single-cell-based clonal evolutionary tree of primary tumors, we refer to the progenitor cells whose descendant population occupies a fraction of the primary greater than $2\alpha$ as "Detectable Ancestors at $\alpha$" (DA$\alpha$). $B_{md}$ is the branch length (measured in variants) from the metastatic seeding cell to its most recent DA at $\alpha$ (MRDA$\alpha$). $B_p$ is the total branch length of the genealogies of DA$\gamma$, after subtracting the branch between the founder and the MRDA$\gamma$ of the seeding cell (Fig 2A). Accordingly, the variants accrued in the DA$\alpha$ are Detectable Variants at $\alpha$ (DV$\alpha$). $B_{md}$ and $B_p$ involve the most recent detectable variants at $\alpha$ (MRDV$\alpha$) and $\gamma$ (MRDV$\gamma$), respectively.

To realize the definitions of $B_{md}$ and $B_p$, we implemented `Comet`, a single-cell-based spatial tumor growth model to simulate virtual primary and metastatic tumor pairs and collect the corresponding virtual sequencing data (see Section 4, and Methods). The phylogenetic distance that defines $B_{md}$ on the evolutionary trees of the virtual primary tumor explains more than 80% of the variance of $B_m$ measured from virtual sequencing data. By contrast, the tumor size (as a surrogate of time) when the seeding cell disseminates does not explain reliably the measured divergence (Fig 3A). Notably, although $B_m$ seems to be positively associated with tumor size when the primary tumor grows rapidly, i.e., with a small death-to-birth ratio ($\rho$ = 0.72) [39] and weak subclonal selection ($s < 0.05$), it displays a characteristic "zig-zag" pattern when the tumor expands under high death rate ($\rho$ = 0.99) [40] and strong selection ($s \geq 0.1$) (Fig 3B). On the other hand, $B_p$ displays little variability when tumor proliferates quickly, whereas it exhibits a "decreasing staircase" pattern when the tumor grows under strong selection (S1(B) Fig). The phylogenetic definition of $B_p$ in the single cell trees explains 90% of the variance of the measured $B_p$ under models involving strong selection (S1(A) Fig). Moreover, the ranges of $B_m$ and $B_p$ become wider as the tumor grows larger. The time-series that depict $B_m$ and $B_p$ across seeding times are heteroscedastic under selective growth modes, i.e., their variability is different across time (Median $p - value < 0.01$ for $s \geq 0.1$, White's test for heteroscedasticity [41], S2 Fig).

**2.1.1 Non-monotonic dependence of M-P divergence on dissemination time.** The zig-zag pattern of $B_m$ suggests that the number of metastasis-specific variants accumulated before dissemination, $B_{md}$, is not monotonically related to the metastatic dissemination time. Assume that seeding of a cell that has $k$ variants results in $B_{md}^k$. We want to calculate $B_{md}^{k+1}$ when this cell stays in the primary tumor and one of its progeny cells, which has just acquired one more variant, seeds the metastasis instead (i.e., the new seeding cell has $k + 1$ variants in total). Assume that the progenitor cell which stays in the primary eventually grows into a subclone that occupies a fraction $f$ of the entire tumor. If $f$ is too small to be detectable ($f < 2\alpha$), $B_{md}^{k+1}$ would increase from $B_{md}^k$ by one. However, if $f$ is large enough to be detectable ($f \geq 2\alpha$), the progenitor cell with $k$ variants would become a detectable ancestor (DA$\alpha$), and so do all of its ancestors. In this case it would be the most recent detectable ancestor at $\alpha$ (MRDA$\alpha$) of the new seeding cell and $B_{md}^{k+1}$ would only refer to the newborn variant (Fig 2B). More generally, the birth of a detectable subclone ensures detectability of the variants harbored by the subclone founder, and when the seeding cell belongs to its progeny, the variants in MRDA$\alpha$ will be subtracted from $k$ to yield the corresponding $B_{md}^k$. Therefore,

$$B_{md}^k = k - V_{\text{MRDA}}(k, \alpha) \tag{1}$$

where $V_{\text{MRDA}}(\cdot, \cdot)$ is a function that takes as an input a given seeding cell with $k$ variants and a

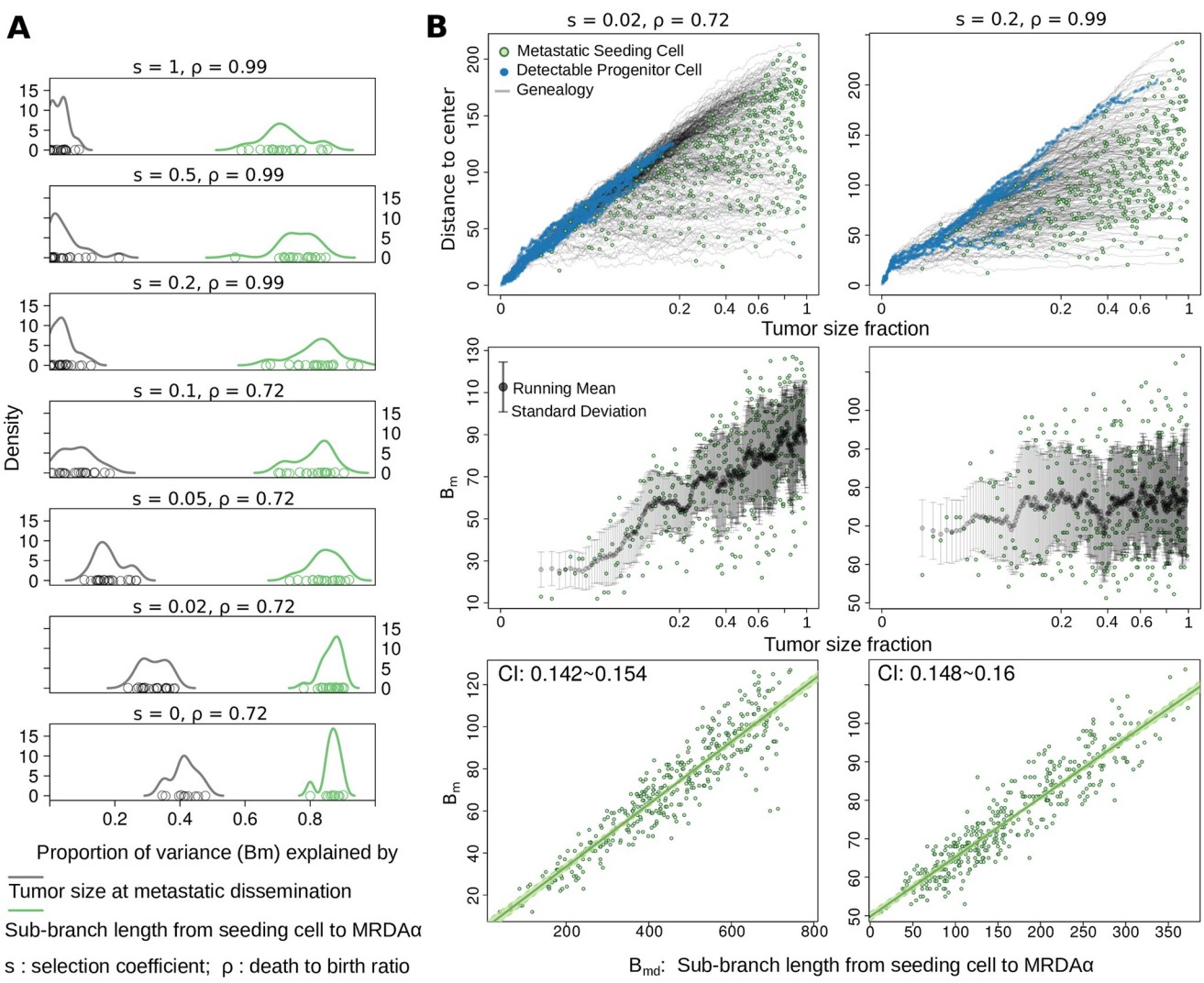

**Fig 3. Single-cell-based spatial simulation of paired metastatic and primary tumors validates the phylogenetic definition of $B_{md}$ and emphasizes that $B_m$ can appear sensitive or insensitive to seeding time based on the kinetics of primary tumor growth. (A)** $B_{md}$, the sub branch length (measured in cell generations) from the metastatic seeding cell to its most recent detectable ancestor (MRDA$\alpha$) captures the majority of the variance of $B_m$ measured from the virtual tumors (green density curves). By contrast, the tumor size at metastatic dissemination, as a surrogate of dissemination time (gray density curves) does not faithfully explain the measured $B_m$. **(B)** Two virtual tumors with distinct kinetics exemplify the conditional dependence of $B_m$ on metastatic dissemination time. Upper panel: the genealogy trajectories (gray lines) of metastatic seeding cells (green dots) along the expansion of the primary tumor in a three dimensional lattice. The Euclidean distance from a cell to the center of the lattice is shown against the tumor size fraction when the corresponding cell disseminates. A cell's distance to the center is strongly correlated with its mutation burden (S3 Fig), reflecting the spatial constraints imposed in our model. The tumor size fraction is plotted at its cube root scale to reflect the clock of actual time. Ancestor cells that are detectable at a frequency greater than 0.01 are marked as blue dots; Middle panel: $B_m$ is plotted against the tumor size fraction when the seeding cell (green dots) disseminates, the running mean (black dots) and standard deviation (gray bars) is also shown; Lower panel: $B_m$ is shown against the $B_{md}$, the sub-branch length from the seeding cell to its MRDA$\alpha$. The slope of the linear regression is consistent with the passenger variant rate (0.15) used in the model, CI: 95% confidence interval.

detectability frequency threshold $\alpha$, and outputs the number of variants in its MRDA$\alpha$. Similarly, if we use $B_p^k$ to express the number of detectable primary-specific variants that are absent in the seeding cell, we have

$$B_p^k = V_{\text{PRIMARY}}(\gamma) - V_{\text{MRDA}}(k, \gamma) \tag{2}$$

where $V_{\text{PRIMARY}}(\gamma)$ denotes the number of variants with frequency above a threshold $\gamma$.

While the total number of variants in the seeding cell indeed increases with dissemination time, the number of detectable ones is heavily influenced by the probability of subclonal expansions; a fact that further complicates the study of the relation between $B_{md}^k$ and $B_p^k$ and seeding time (Fig 2C).

**2.1.2 Mathematical insights on the expected $B_{md}$ patterns.** Let $d_{c_j}$ denote the probability variant $c_j$ ends up being detectable in the primary tumor. In Methods we show that under the infinite allele model, the probability $B_{md}^k$ takes the value $i$ can be expressed in a surprisingly simple form in terms of $d_{c_j}$, as

$$\Pr\left[B_{md}^k = i\right] = d_{c_{k-i}} - d_{c_{k-i+1}}. \tag{3}$$

Eq (3) essentially relates the length of $B_{md}$ with the probabilistic decay in detectability along the lineage of the seeding cell (Fig 4C). Parameter $k$ refers to the number of variants that are present in the seeding cell, effectively acting as a clock for the dissemination time. Calculating $B_{md}^k$ for different values of $k$ enables us to reason about the expected $B_{md}$ under settings of early (small $k$) or late (large $k$) dissemination. It also allows us to reason about the monotonicity of

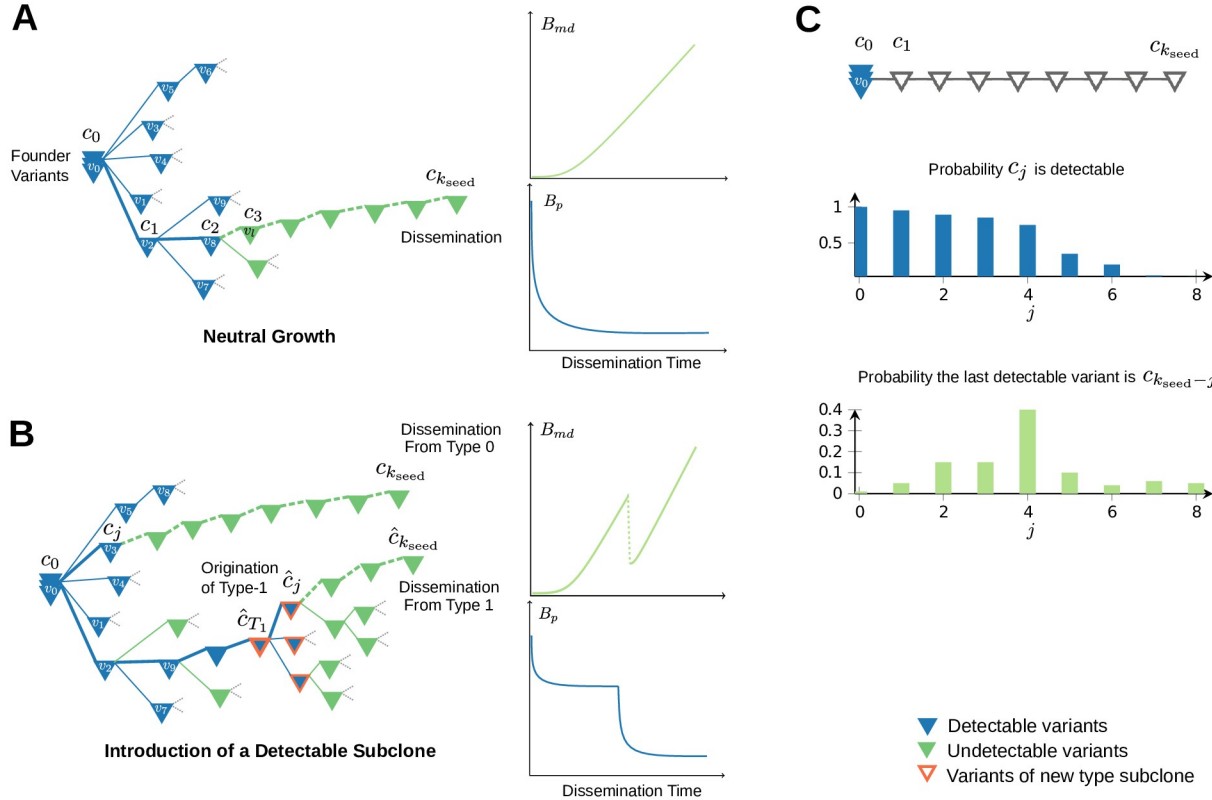

**Fig 4. Characteristic $B_{md}$ and $B_p$ patterns.** (**A**) Example phylogenetic tree of somatic variants under neutral model of growth. Notice that the expected $B_{md}$ time series are increasing with seeding time since the number of detectable variants along a general branch are expected to reach a fixed limit, beyond which new variants are almost surely undetectable. Similarly, the expected $B_p$ patterns are decreasing (cf. (8)) up until a point where they stabilize to a fixed small number. (**B**) Example tree depicting the introduction of a new type of cells. Notice that $B_{md}$ at time $k_{\text{seed}}$ is radically different when seeding originates from type-0 as opposed to type-1. Assuming that seeding is proportional to the fitness of the types, the expected $B_{md}$ time series are expected to drop abruptly for seeding times right after the introduction of the new type. Similarly $B_p$ tracks these drops. (**C**) The connection between the decay in detectability of the variants along the seeding cells lineage and the distribution of the $B_{md}^{k_{\text{seed}}}$ lengths. Notice that a sudden drop in the detectability probabilities along this path, leads to a spike in the probability of the corresponding length of $B_{md}^{k_{\text{seed}}}$.

the sequence $\{B_{md}^k\}_{k=0,1,\dots}$, which is crucial for accurate estimation of seeding time, based on specific genomic divergence measurements. In particular, notice that a strictly increasing $\{B_{md}^k\}_{k=0,1,\dots}$ would allow, in principle, to unambiguously relate divergence measurements with seeding times; on the contrary, if the monotonicity of $\{B_{md}^k\}_{k=0,1,\dots}$ is not guaranteed, an unequivocal interpretation of generic divergence measurements cannot be guaranteed in advance—at least not without further assumptions regarding the growth dynamics of the tumor under study, or additional information that could allow one to break this ambiguity.

To further explore the implications of Eq (3) in the analytically tractable case of neutral growth, building on the results of [29], we derive an approximate form for the expected $B_{md}$ as a function of the seeding time $K$, the detectability threshold $f$, the mutation rate $u$, and the death-to-birth ratio of the tumor cells $\rho$

$$\mathbb{E}[B_{md}^{\text{neutral}}; K, f, \rho] =$$
$$= \sum_{j=1}^{K} j \left[ \left( \frac{u}{u - \log(1 + (\rho - 1)f)} \right)^{g(c_{K-j})} - \left( \frac{u}{u - \log(1 + (\rho - 1)f)} \right)^{g(c_{K-j+1})} \right] \tag{4}$$

where $g(\cdot)$ is an approximate mapping that associates the variants that are particular to the seeding cell, with their absolute order of appearance in the overall primary tumor (specified in Section 4.1.2).

Importantly, we find that under neutral growth the resulting sequence $\{\mathbb{E}[B_{md}^{\text{neutral}}; K, f, \rho]\}_{K=0,1,\dots}$ is in fact monotonically increasing with time (Fig 4A). In particular, one can discern two distinct phases, the shape of which is modulated by the parameters of the model; an initial phase where the increase in expected $B_{md}$ is almost imperceptible (the length of this phase is controlled by $\rho$); and a later phase of steady increase in expected $B_{md}$ at almost constant rate (that is not particularly sensitive to $\rho$). S5 Fig demonstrates the related patterns. Observe that larger values of $\rho$ are indeed associated with larger periods of time for which $B_{md}$ is expected to be very small. However, invariably, as more and more variants are being introduced in the primary tumor, their detectability probability plummets, thereby leading to a predictable increase in expected $B_{md}$ in late seeding settings. This is a direct consequence of the exponential decay in detectability probabilities of newly emerging variants down the phylogenetic tree. Intuitively, when seeding happens late under neutral growth, the later accumulating variants on the seeding cell are almost surely undetectable, and the depth of the MRDA$f$ converges to a fixed value that is no longer affected by $K$.

The situation, however, is markedly different in the non-neutral growth setting. To see this consider the case where a new type of advantageous cells is introduced to the primary tumor at time $T_1$. Also assume that the overall population of cells that carry the beneficial variant in the primary tumor (the one that characterizes the advantageous type) reaches a fraction $f_1$. When dissemination happens at time $k < T_1$ the seeding cell originates necessarily from the original type of cells, and thus, the above reasoning still applies subject to a modification of the detectability threshold which now needs to account for the fact that the relative frequency that a variant needs to achieve to become detectable in the overall tumor at detection time is now $f/(1 - f_1)$. On the other hand, if dissemination happens at time $k > T_1$, the value of $B_{md}$ will be predominantly determined by whether the seeding cell carries the beneficial variant or not. Indeed, if the seeding cell originates from the beneficial subpopulation, the expected $B_{md}$ will be inevitably smaller since only variants that are introduced after the beneficial one (variants arising after time $T_1$) could end up being undetectable. The detectability of the beneficial type itself ensures such outcome (Fig 4B).

Thus, to reason about the effect of the introduction of new types on the expected $B_{md}$, we need to explicitly account for the probability the seeding cell originates from the subpopulation of advantageous cells as a function of seeding time, $k$. We denote such probability $\phi_1^k$ (similarly we use $\phi_0^k = 1 - \phi_1^k$ to denote the probability of seeding from the original type of cells), and express $\{\mathbb{E}[B_{md}; k]\}_{k=1,\dots}$ as

$$\mathbb{E}[B_{md}; k, T_1, f_1] \approx$$
$$\approx \phi_1^k \times \mathbb{E}\left[B_{md}^{neutral}; k - T_1, \frac{f}{f_1}, \rho_1\right] + \phi_0^k \times \mathbb{E}\left[B_{md}^{neutral}; k, \frac{f}{1 - f_1}, \rho_0\right]. \tag{5}$$

In Methods we explore several different choices of $\{\phi_0^k\}_{k=1,\dots}$, $\{\phi_1^k\}_{k=1,\dots}$ that can capture different assumptions regarding the underlying seeding potential of the different types of cells that are present in the population. Notably, our findings suggest that the monotonicity of the sequence $\{\mathbb{E}[B_{md}; k]\}_{k=1,\dots}$ is expected to break by the introduction of newly emerging advantageous types. Such effect is unavoidable when dissemination capability is positively associated with fitness advantage, however we find that the expected drop in $B_{md}$ remains significant even when seeding happens uniformly at random from all living cells.

## 2.2 Single-cell spatial tumor growth simulation reveals that dependence of M-P divergence on seeding time is conditional on the underlying modes of tumor growth

To corroborate the mathematical insights in a spatial growth setting, we develop Comet; a state-of-the-art simulator of paired primary and metastatic tumor growth. From a technical point of view Comet extents the single cell-based spatial tumor growth model [42]. Various parameter settings of driver genomic variants and cell death-to-birth rates enable our model to encompass three distinct and representative tumor growth dynamics: progressive diversification, branching, and linear evolution (see Methods, Table 1). We imposed strict spatial constraints to reflect the patterns of peripheral growth (S3 Fig) which were observed in solid tumors [43]. In addition to driver events, we simulate passenger variants that occur at a given rate $u$, and record variant lineages as the tumor expands. To track the changes of M-P divergence along with the expansion, our model allows monoclonal dissemination and metastatic seeding as the population size of the primary tumor increases (every 25,000 cells). Both virtual primary and metastatic tumors reach a final volume of $10^7$ cells [19]. We perform multi-region

**Table 1. Parameters and settings in Comet, the single cell based spatial computational modeling of evolving tumors.**

| Parameter | Value | Meaning |
|---|---|---|
| $\lambda$ | 0.25 | initial birth rate per unit time |
| $\mu$ | 0.2475, 0.18 | intial death rate per unit time |
| $\rho$ | 0.99, 0.72 | the ratio between the initial death and birth rate |
| $\psi_i$ | stochastic | proportion of empty neighboring sites of the $i$-th cell |
| $m_i$ | 0 | Intra-tumoral migration rate |
| $u$ | 0.15 | passenger variant rate |
| $u_b$ | $1 \times 10^{-5}, 5 \times 10^{-5}, 5 \times 10^{-4}$ | beneficial variant rate |
| $s$ | 0, 0.02, 0.05, 0.1, 0.2, 0.5, 1 | mean selection coefficient |
| $s_d$ | a half of the corresponding $s$ | standard deviation of selection coefficient |
| $N_s$ | 25,000 (12,500, 50,000) | tumor size increment to trigger seeding |
| $N$ | $1 \times 10^7$ | final tumor size |

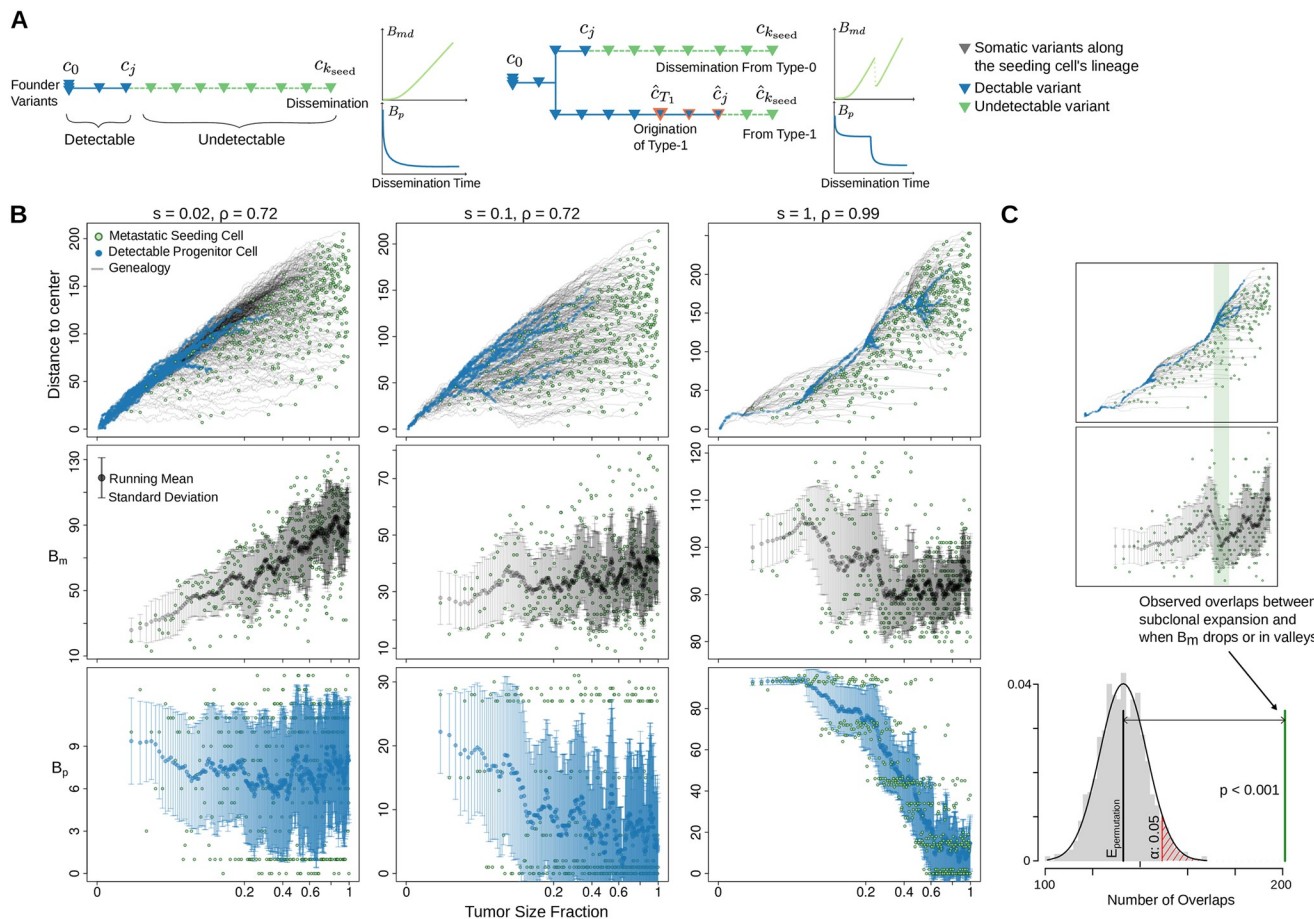

**Fig 5. Spatial computational model verifies that the growth mode governs the dependence of M-P divergence on seeding time.** (**A**) Mathematical models illuminate the expected patterns of $B_{md}$ and $B_p$ under neutral (left) and selective dynamics (right, corresponding to the second and third columns of panel **B**), respectively. Detailed example plots of mathematical analyses can be found in Fig 7 and S1 Appendix. The variant genealogy schema of seeding cells is also shown. (**B**) Virtual tumors with three representative clonal kinetics and the changing patterns of M-P divergence along with the metastatic seeding time. Upper panel: the genealogy trajectories (gray lines) of metastatic seeding cells (green dots) during the expansion of the primary tumor in a three-dimensional lattice. The Euclidean distance from a cell to the center of the lattice is shown against the fraction of the final primary tumor size when the corresponding seeding cell is born and disseminated. A cell's distance to the center is strongly correlated with its mutation burden (S3 Fig), reflecting the spatial constraints imposed in our model. The tumor size fraction is plotted at its cube root scale to reflect the clock of actual time. Blue dots represent progenitor cells that are detectable at a frequency higher than 0.01. Middle panel: $B_m$ is plotted against the tumor size fraction when the seeding cell disseminates, where black dots and gray bars indicate the running mean and standard deviation of $B_m$, respectively. Lower panel: change of $B_p$ with different seeding time, blue dots and bars represent the running mean and standard deviation of $B_p$, respectively. (**C**) The subclonal expansion occurrence is strongly associated with drops and valleys of $B_m$ in spatial simulations (see Methods).

sampling of the virtual tumors and calculate the variant allele frequencies by assuming a negative binomial distribution of the sequencing depth [44] for each sample. $B_m$ and $B_p$ is measured from the virtual sequencing data accordingly.

Our model makes it possible to study both subclonal diversification and subclonal convergence in the spatial tumor growth setting (Figs 5 and 6A). We note that while the parameters used are model-dependent, the clonal evolutionary patterns realized by different levels of positive selection and death-to-birth ratio are compatible with those caused by other evolutionary and ecological forces. For example, competition between tumor and normal cells, which is not modelled here, can amplify clonal selection [13]. Weak selection ($s < 0.05$) coupled with small death-to-birth ratio ($\rho = 0.72$) leads to a pattern of progressive diversification which has been discussed previously [45]. Interestingly, increasing the selection coefficient further facilitates

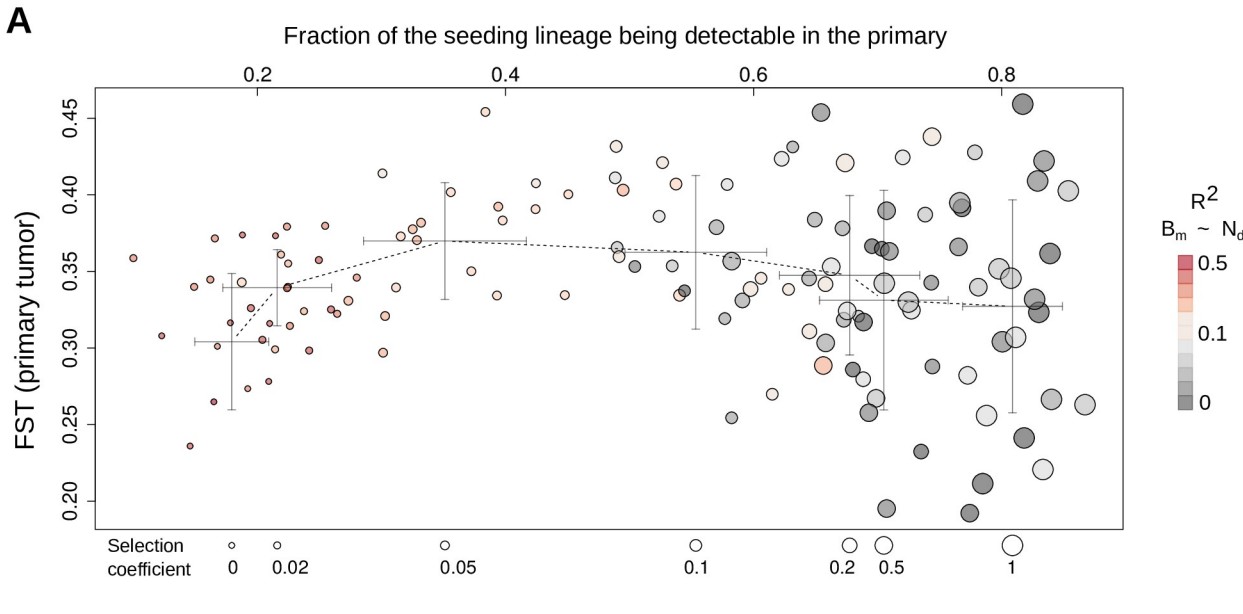

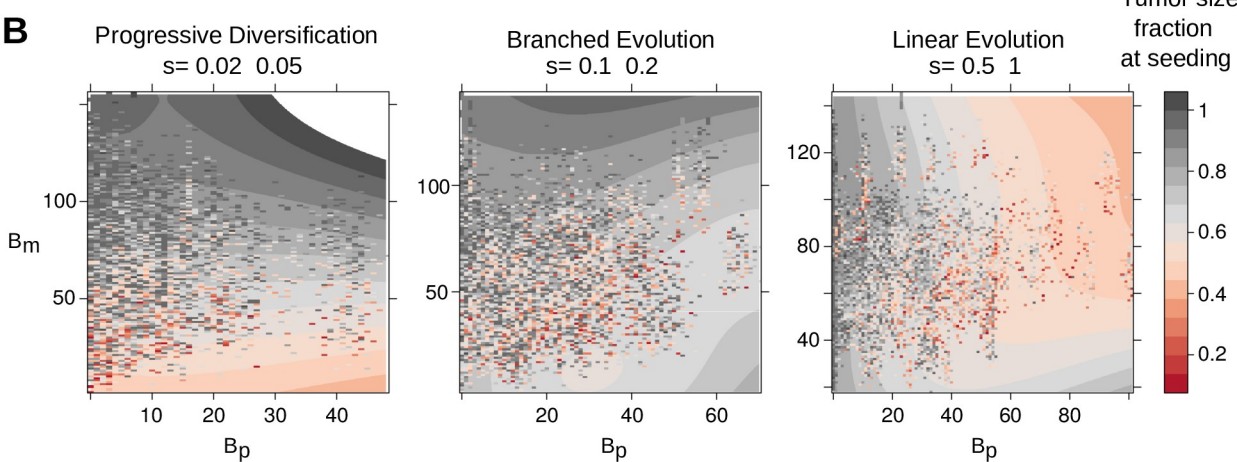

**Fig 6. Summary of the virtual tumor dynamics and the mapping between seeding time and M-P divergence.** (**A**) For each simulation (a circle), we calculated the average pair-wise Fst (Fixation index) of ten randomly sampled regions of the virtual primary tumor, to reflect the between-region genetic divergence. We plotted the Fst against the average fraction of variants of the seeding cell that fell above the detection limit in the primary tumor. The size of the circle increases with the selection coefficient. For each selection level, the crossbars show the standard deviation along the two axes. Color scale indicates the proportion of variance of $B_m$ explained by the seeding time (based on 10-quantile of all the values). (**B**) We group the various simulations into three representative clonal dynamics: progressive diversification ($s \leq 0.05$), branched evolution ($s = \{0.1, 0.2\}$) and linear evolution ($s \geq 0.5$). For each group, we plot the actual seeding time (color hues, using the fraction of the final primary tumor size at metastatic dissemination as a surrogate) on top of the corresponding $B_m$ and $B_p$ values. A smoothed layer shows the general distribution of the seeding time.

late subclonal sweeps at different scales (Fig 5B, upper panel). Moderately strong selection ($s = 0.1$) leads to a branched evolution pattern where a few detectable subclones jointly constitute the whole tumor. High death-to-birth ratio ($\rho = 0.99$) coupled with strong selection ($s = 1$) introduces a linear growth pattern, where advantageous subclones broadly sweep the population from time to time and dominate each "wave" of growth.

The results of the spatial tumor growth model are aligned with our mathematical insights (Fig 5), suggesting that the non-monotonicity of M-P divergence is ubiquitous in representative

tumor growth models, regardless of the specific parametric settings. When the clonal composition of the primary tumor progressively diversifies, $B_m$ increases with seeding time (Fig 5B, left column), and $B_p$ remains small. However, $B_m$ starts to become less sensitive to metastatic dissemination time when the tumor grows under deep branching evolution (Fig 5B, middle column). Under linear growth dynamics, $B_m$'s non-monotonic relationship with seeding time is evident (right column). Notably, the mathematically predicted zig-zag pattern of the expected $B_{md}$ is evident in all virtual tumors. Consistent with results from the non-spatial mathematical modeling, the subclonal expansion in the virtual primary tumors significantly co-occurs with the drop of $B_m$ (Fig 5C). This consistency is also strongly supported by the fact that $B_{md}$, measured from the genealogy of the seeding cells, explains $B_m$ much better than the seeding time (Fig 3). Under the dynamics of clonal convergence, $B_p$ has the tendency to transit to lower levels if the seeding cell disseminates later. The "decreasing staircase" pattern of $B_p$ is particularly evident for the linear growth mode.

In the virtual primary tumors, the between-region genetic divergence (measured by the Fixation index [46]) initially increases with the selection coefficient when $s < 0.05$, in agreement with our previous findings using a deme-based model [18]. However, higher and more influential selection levels ($s \geq 0.1$) reduce the genetic diversity in the primary tumor and create the dynamics of subclonal convergence (Fig 6A and S4 Fig). In the meantime, more variants of the seeding cell become detectable in the primary tumor on average (i.e., less variants contribute in $B_m$) as we raise the selection coefficient. When $s \geq 0.1$, more than 50% of the seeding cell lineage on average is detectable in the primary tumor, and $B_m$ starts to become much less sensitive to seeding time (Fig 6A).

In summary, the metastatic seeding time is a radically different function of $B_m$ and $B_p$ under distinct dynamics of primary tumor growth (Fig 6B). Under the mode of progressive diversification ($s \leq 0.05$), $B_m$ is the most informative indicator. For linear evolution ($s \geq 0.5$), the associated step-wise drops in $B_p$ may be leveraged to inform the timing. Under branched evolution, however, the situation is significantly more perplexed; large $B_m$ and $B_p$ readings point to late seeding, whereas mid-range readings suggest a non-trivial—possibly non-linear—relationship between $B_p$, $B_m$ pairs and metastatic dissemination time. These patterns are reproducible under different values of parameter $N_s$ (S9 Fig). Furthermore, the non-monotonicity of M-P divergence over time leads to increased uncertainty in mapping certain pairs of $B_m$ and $B_p$ onto metastatic initiation times (variegated pattern, Fig 6B). In the latter setting, additional indicators may be necessary to allow an accurate statistical reconstruction of the metastatic dissemination chronology.

## 3 Discussion

To reliably bridge the scales between the divergence at the genomic level and the metastatic spread at the cellular level, we focus on if and how dissemination time and tumor growth dynamics remold the between-tumor heterogeneity. The major findings are (1) Depth of most recent detectable variant characterizes the M-P divergence; (2) Decay in the probability of a variant being detectable by sequencing determines the expected observable M-P divergence; (3) Dissemination from late detectable subclones leads to an abrupt drop in M-P divergence and (4) Spatial model verifies that the growth mode governs the dependence of M-P divergence on seeding time.

Our results pave the way towards building effective statistical models that connect M-P divergence to a metastatic timeline. Firstly, the growth-dependent relationship between $B_m$, $B_p$ and seeding time indicates that knowing the primary tumor growth dynamics is the prerequisite for translating M-P divergence. For example, a small $B_m$ and $B_p$ [1, 4] may suggest early

seeding for a tumor grown under progressive diversification, but late dissemination under linear evolution (Fig 6B). For neoplasms characterized by deep branching evolution, the timing of metastatic spread may be more difficult to identify. This is particularly relevant for tumors showing invasive glandular structure, which have been associated with deep branching mode [13]. Secondly, the analytically predicted—and computationally verified—$B_m$, $B_p$ patterns indicate that more elaborate statistical models, that take into account non-monotonic dependencies, are necessary to utilize such measurements effectively. Lastly, the difference between $B_m$ and $B_p$, with the latter also being measured based on a lowered substantial presence threshold ($\alpha$), in theory cancels out the drop effect upon the introduction of a new detectable subclone and may inform better the seeding time. However, this quantity can be heavily affected by the noise in variant detection at low frequencies. As the M-P divergence alone, may not always carry enough information to accurately infer the seeding time (under branching growth mode in particular), additional indicators, as well as other clinical information may be leveraged to increase the confidence of the inference [47].

We mathematically showed that the M-P divergence also reflects the interplay between clonal dynamics and the subclonal preference of seeding [2], which may imply the genetic determinants of metastatic dissemination [48]. Based on our discovery, one may quantitatively test hypotheses about such determinants using multi-region sequencing of multiple metastases and paired primary tumor of the same patient. For example, always seeding from a rare subclone that has metastatic-specific driver variants may lead to a $B_m$ that is more monotonically related to seeding time compared to planting from a major subclone which appears late and possesses a growth advantage. Moreover, the step-wise decrease of $B_p$, if seen in patients with multiple metastases, could suggest an increased probability of seeding from actively growing subclones [49, 50]. On the other hand, the actual pattern of M-P divergence could also hint on an evolutionary bottleneck caused by microenvironmental factors. For example, seeding cells that carry fewer unique variants may experience a safer passage to distant sites, by reducing the chance of being eliminated by immune surveillance [11].

The measured M-P divergence from Next Generation Sequencing is not only determined by evolutionary forces but also affected by the detection regime. Usually, only a handful of samples of each lesion are collected for sequencing [5, 51], and their spatial location in the corresponding tumor is either unknown or recorded with poor resolution. Depending on the sampling location and clonal structure of the tumors, $B_m$ and $B_p$ can be artificially enlarged or reduced. For example, if we reduce the number of virtual samples from ten to five, while the effect of evolutionary determinants is still evident, the same M-P divergence could map to a different seeding time due to the change in sampling alone (S10 Fig). In addition, primary tumor samples often come from formalin-fixed, paraffin-embedded tissues that have been stored for much longer than the paired metastatic samples, leading to unequal DNA quality between the two [52]. Between-sample heterogeneity can be further over- or under-estimated due to low tumor purity. As the ploidy and purity are often jointly estimated [53], the copy number evolution in the metastatic progression of certain tumor types [54, 55] may confound the quantification of genomic divergence between samples. Therefore, an assessment of the variability in heterogeneity due to suboptimal detection is necessary before comparing the real data with computational models, and is thus, beyond the focus of this study. Given the complementarity between bulk longitudinal sampling and single-cell sequencing, we envision that some hypotheses generated by our models may be better tested by combining these two sequencing data sources, as they provide the ultimate resolution in distinguishing the mode of clonal dynamics and tracking the genealogies among tumor cells.

In our framework, time zero is set at the birth of the tumor founder cell, for which the full metastatic competence may not necessarily have been bestowed. We make minimal

  

assumptions about the determinants of seeding potential by only requiring that it is the progeny cell of the founder that spreads, owing to genetic or non-genetic factors. If one assumes that the founder is a fully transformed (malignant) cancer cell [1], which grows under progressive diversification *in vivo* [16], early seeding is more likely given a small M-P divergence. However, this is circular reasoning as the starting cell already has the full potential of metastasis, and short waiting time before seeding is not surprising. Another related consideration is that the actual primary tumor size, as a surrogate of time, should be determined by all leaves originated from the founder cell, not just those of the more malignant subclone [56]. Our model also considers several distinct but representative growth dynamics of the founder cell's expansion. This flexible setting helps us to chart the landscape of connections between M-P divergence and metastatic seeding times.

Since this is the first attempt to analytically characterize M-P divergence patterns, we focus on key but inevitably simplified considerations; nevertheless, our framework can serve as a foundation for future data-driven efforts aiming at understanding more complex metastatic dynamics. The actual mode of primary tumour growth may vary when the physiological conditions of the micro-environments vary, resulting in complex dynamics that our model does not capture. For example, clonal competition could increase, and the tumour may slow down in terms of growth on account of crowding over the available resources [57]. As a result, M-P divergence generated by a real patient tumor is even less likely to be monotonic as the tumor expands. On the other hand, while we do leave the dynamics of metastatic tumor unexplored by focusing on $B_{md}$, the modes of the primary tumor can also be applied to the secondary tumor. We expect that the length of $B_m$ rather than $B_p$ is affected by the growth pattern of metastatic tumor given its fitness in the distant site and possible iatrogenic perturbations [58]. We analyzed monoclonal seeding solely, which accounts for the majority of distant metastasis [35], but the analytical thinking is extendable to reason about more complex poly-clonal seeding [59, 60]. Poly-clonal seeding (or multi-cell seeding, under the strict definition of a clone) involves the following two scenarios: monophyletic (genetic homogeneity among the seeding cells) and polyphyletic (genetic heterogeneity instead) [5]. Our framework can be applied directly to monophyletic seeding (note that in terms of the resulting M-P divergence, seeding from a single cell is genomically equivalent to the colonization by a group of cells in the monophyletic lineage). In contrast to monophyletic dissemination, polyphyletic seeding would lead to a reduction in M-P divergence as more genetic diversity in the primary is transmitted to the metastasis (e.g., suggested by subclonal sharing between primary and metastasis sequencing data [60], assuming that purity and local copy number differences are taken into account). Nevertheless, the tumor growth mode and seeding tempo are still expected to regulate the observed M-P divergence. For example, seeding simultaneously as a cluster, or consecutively could lead to distinct patterns in M-P divergence [61]. Our framework can be used to study the divergence between the primary and each seeding subclone characterised from sequencing data [62] but more specialized considerations are required to fully model the polyphyletic seeding. Moreover, here we do not model the circulatory phase of metastasis, a bottleneck that could confound the M-P divergence further. Unified mathematical models have been proposed to understand the full picture of this phase [36, 37, 63], which would aid the decomposition of the complexity of metastases. While our analysis here is applied to the specific case of metastatic progression, the usefulness of our approach extends to the case of comparing longitudinal tumor samples, as well. Finally, our findings emphasize that investigating tumor growth dynamics in detail has important analytical purchase, particularly for the efforts that seek to translate heterogeneity among longitudinal samples to the evolutionary timeline of progression.

  

## 4 Methods

### 4.1 Mathematical modeling

**4.1.1 Tree-based characterization of $B_{md}$ and $B_p$.** Consider the phylogenetic tree of the surviving variants in the primary tumor, i.e., the tree rooted on the variant that characterizes the founder cell, as well as the additional variants that are accumulated in the primary tumor as it evolves over time. Let us index such variants according to their order of appearance in the primary tumor, as $\mathcal{V} = (v_0, v_1, \ldots, v_{k_{\text{primary}}})$. At a certain point in time a cell departs from the primary tumor and becomes the founder cell of a metastatic tumor. We call this event, metastatic seeding, and we refer to the time this event takes place, as *seeding time*; henceforth denoted as $k_{\text{seed}}$. Notice that the variants that are present on the seeding cell, $\mathcal{C}$, at seeding time, are by definition, also present in the primary tumor, i.e., $\mathcal{C} \subset \mathcal{V}$. However, the final frequencies reached by a certain subset of such variants upon observation of the primary tumor, can potentially fall below the detectability threshold of current sequencing technology. This implies that only a subset of the variants, $\mathcal{C}_{det} \subset \mathcal{C}$, will be registered, in practice, as being part of the primary tumor. On the other hand, all such variants will be detectable on the metastatic tumor, since they will be present in 100% of the metastatic cells as part of the founder of the metastasis. Here we are interested in characterizing the number of variants in $\mathcal{C}$, that end up not being detectable in the primary tumor, $B_{md}^k$, as well as the number of frequent variants in $\mathcal{V}$ that are absent from the metastatic tumor, $B_p$.

Let $c$ be a seeding cell and also let $\mathcal{C} = (c_0, c_1, \ldots, c_{k_{\text{seed}}})$ be its associated set of variants, indexed according to the order of appearance in the tumor. Let $d_{c_j}$ denote the probability variant $c_j$ is detectable in the primary tumor, and also let $B_{md}^k$ be the random variable that counts the number of variants in $\mathcal{C}$ that are not detectable in the primary tumor at observation time. Under the infinite allele model if a seeding-cell specific mutation $c_j$, becomes detectable in the primary the same will be true for all $c_i < c_j$. Thus, $\{d_{c_j}\}_{j=0,1,\ldots}$ will be a decreasing sequence that starts from $d_{c_0} = 1$ and goes to 0 with $j$. Notice that for $B_{md}^k$ to take the value $j$, the final frequencies of variants $c_{k_{\text{seed}}}, c_{k_{\text{seed}}-1}, \ldots, c_{k_{\text{seed}}-j+1}$ are necessarily below the detectability threshold, while $c_{k_{\text{seed}}-j}$ is above. This leads to the probability distribution

$$\Pr[B_{md}^{k_{\text{seed}}} = k_{\text{seed}} - j] = \begin{cases} d_{c_j} - d_{c_{j+1}}, & j < k_{\text{seed}} \\ d_{c_{k_{\text{seed}}}}, & j = k_{\text{seed}} \end{cases} \tag{6}$$

For a detailed derivation of (6) see S1 Appendix.

We now turn our attention on $B_p$. Let $\mathcal{M}$ be the set of variants that reach a frequency of at least $\gamma < f_P < 1$ in the primary tumor. By definition we have

$$B_p^{k_{\text{seed}}} = |\mathcal{M}| - V_c^{k_{\text{seed}}} \tag{7}$$

where $V_c^{k_{\text{seed}}} = |\mathcal{C}_{k_{\text{seed}}} \cap \mathcal{M}|$ is the number of variants of the seeding cell that are also included in $\mathcal{M}$. Taking into consideration the phylogenetic view detailed above, in conjunction with the definition of $\mathcal{M}$, reveals that $V_c^{k_{\text{seed}}}$ is, in fact, equivalent to $V_{\text{MRDA}}(k, \gamma)$. This allows us to express (cf. (1)) the sought quantity (also explicitly considering the reference frequency $\gamma$) as

$$B_p^k(\gamma) = |\mathcal{M}| - k + B_{md}^k(\gamma) \tag{8}$$

In practice, $|\mathcal{M}|$ can be inferred from multi region sampling of the primary tumor at detection time [14] and is, therefore, assumed known for the scope of our analysis. Importantly, Eq (8) reveals that $B_p^k(\gamma)$ decays linearly with seeding time, $k_{\text{seed}}$, while it is positively

correlated with $B_{md}^{k_{\text{seed}}}(\gamma)$. Intuitively this suggests that sudden drops in the sequence $\{B_{md}^k(\gamma)\}_{k=0,1,\dots}$ are expected to be mirrored by similar drops in $\{B_p^k(\gamma)\}_{k=0,1,\dots}$, whereas ranges where $\{B_{md}^k(\gamma)\}_{k=0,1,\dots}$ is increasing linearly with time, would be mapped to $B_p^k(\gamma)$ subsequences that remain constant.

Having characterized the general form of $B_{md}^k$ and $B_p^k$, we will now investigate how can the $B_{md}^k$ probability mass function can be approximated for the analytically tractable case of the exponential growth model without spatial structure. We first restrict our attention to the case of neutral growth, while also considering the accumulation of passenger variants that may become detectable in the observed tumor. We will then proceed to the investigation of the effects of the introduction of advantageous types of cells in the population.

**4.1.2 $B_{md}^k$ under neutral growth.** Following [29], we model tumor growth using a multitype branching process that starts with a single type-0 cell. The cells divide with rate $\lambda$ and die with rate $\mu$. Upon each division, one of the daughter cells acquires a new passenger variant with probability $u$, which denotes the birth of a new type. Such type, while not being differentiated in terms of $\lambda$ and $\mu$ from type-0, it is uniquely characterized by the newly-acquired genetic variant. To emphasize the fact that from a fitness point-of-view such new types are not advantageous with respect to type-0, we henceforth refer to them as *subtypes* of type-0. Any new variant that appears in the population can be lost due to stochastic fluctuations. The probability that its lineage will not survive is $\rho = \mu/\lambda$, the ratio of the death and the birth rates.

Now consider a seeding cell $c$ and let $(c_0, c_1, \dots, c_{k_{\text{seed}}})$ be the set of variants that it carries, indexed according to their order of appearance in the tumor as above. Within the aforementioned tumor growth setting we are interested in deriving an approximation of the probability any given variant that is present in the seeding cell will be detectable in the final primary tumor. In particular, we are interested in an expression that also takes into account the limitations of sequencing technology. Therefore, we assume that in order to be detectable a variant needs to be present in a fraction of cells of the overall population larger than $f = 2\alpha$. [29] established that the final frequencies of passenger variants in the population is primarily affected by the order by which these variants appeared in the tumor. Therefore, a necessary step towards obtaining the sought approximation utilizing the results of [29] is to derive a mapping between the relative ordering of the variants that are particular to the seeding cell, to their absolute order of appearance in the overall tumor population (see example Fig 4).

Note that such mapping is exactly specified by the topology of the resulting phylogenetic tree that arises by a realization of the aforementioned branching process. It is, therefore, inherently random in itself. However, for the purpose of our analysis, opting for simplicity, we treat this mapping as deterministic, utilizing basic properties of branching processes with exponentially distributed lifetimes. Concretely, consider the lineage of the seeding cell $c$, i.e., the path in the phylogenetic tree, from the root (founder variants, i.e., $c_0$) to the leaf (the last appearing variant, i.e., $c_{k_{\text{seed}}}$). The number of nodes in this path is by definition $k_{\text{seed}} + 1$ and they depict the different variants that are present on the seeding cell $c$. Now, let's associate each node with a positive real number referring to the time of appearance of the corresponding variant in the population. Due to the assumption of exponentially distributed waiting times between cell divisions, it follows that the time difference between the appearance of two consecutive variants along this path is going to also be exponentially distributed with rate $u \times \lambda$. Thus, the distribution of the time needed for variant $c_j$ to appear in the population, can be expressed as the sum of $j$ independent exponentially distributed random variables with identical rates equal to $u \times \lambda$. Therefore the successive expected times of appearance of the variants along the seeding

cell's lineage are given by $t_0 = 0$ (by assumption the time starts when the founder cell is born), and $t_j = \frac{j}{u \times \lambda}$, for $j = 1, \ldots, k_{\text{seed}}$.

Now, the expected number of cells in the population at a given time (conditioned on long-term tumor survival) can be calculated (see e.g., [2]) as

$$N(t) = \frac{1}{\rho} e^{\lambda t} - \left(\frac{1}{\rho} - 1\right) e^{-\lambda t}. \tag{9}$$

Thus, substituting the expected times of appearance of the variants of the seeding cell, $t_j$, to Eq (9) we can get an estimate of the expected number of cells in the population, at the time each of the seeding-cell-specific variants appeared. Utilizing also the fact that the expected number of passenger variants with surviving lineage that are present in a population of $N$ cells is $N \times u$ [64], we derive the final expression that maps the $j$-th seeding-cell-specific variant, $c_j$, to its absolute ordering in the overall tumor population, $v_{g(c_j)}$:

$$g(c_j) = u\left(e^{\left(\frac{\rho}{u} - \frac{1}{u}\right)j} + 2\frac{\sinh\left(\left(\frac{1}{u} - \frac{\rho}{u}\right)j\right)}{\rho}\right). \tag{10}$$

Having obtained the approximate mapping $g(\cdot)$, we can utilize the results of [29], and express the sought probability $d_{c_j}$ as

$$
\begin{aligned}
d_{c_j} &= \Pr\left[\text{the frequency of cells that carry variant } c_j \text{ is at least } f\right] \\
&\approx \left(\frac{u}{u - \log(1 + (\rho - 1)f)}\right)^{g(c_j)}.
\end{aligned} \tag{11}
$$

Hence, the final expression of the expected $B_{md}^k$, is specialized for this setting, as

$$
\begin{aligned}
\mathbb{E}[B_{md}^{\text{neutral}}; K, f, \rho] &= \\
&= \sum_{j=1}^{K} j\left[\left(\frac{u}{u - \log(1 + (\rho - 1)f)}\right)^{g(c_{K-j})} - \left(\frac{u}{u - \log(1 + (\rho - 1)f)}\right)^{g(c_{K-j+1})}\right].
\end{aligned} \tag{12}
$$

$\mathbb{E}[B_{md}^{\text{neutral}}; K, f, \rho]$ and $\text{Std}[B_{md}^{\text{neutral}}; K, f, \rho]$ over seeding times for different values of $\rho$ and $f$ can be found in S5 and S6 Figs.

**4.1.3 On the effects of the introduction of advantageous types.** We study how the introduction of advantageous types affects the monotonicity of $\{B_{md}^k\}_{k=0,\ldots,k_{\text{seed}}}$. Our approach is based on a straightforward generalization of the neutral-growth setting, and our analysis assumes phylogenetic independence between the passenger variants accumulated on the original type-1 cell, and is also conditioned on explicit knowledge of the final population frequencies achieved by the different beneficial variants (in practice, the subclonal architecture of a tumor can be inferred from deep sequencing data [65, 66]).

For simplicity we consider the case where there are two types of cells in the population: (a) type-0 cells, which are characterized by death-to-birth ratio $\rho_0$ and are originated at time $T_0 = 0$ (i.e., type-0 is the type of the founder cell); and, (b) type-1 cells, which are characterized by death-to-birth ratio $\rho_1 < \rho_0$ and are originated in the population at time $T_1$, reaching a final frequency $f_1$ at the time of detection. When seeding happens at time $k$, the seeding cell originates from the type-1 cell population with probability $\phi_1^k$ and from the type-0 cell population with probability $\phi_0^k = 1 - \phi_1^k$. Of course, $\phi_1^k = 0$ for $k < T_1$, since there exist no type-1 cells in the population. Note that different choices of $\{\phi_0^k\}_{k=1,\ldots}, \{\phi_1^k\}_{k=1,\ldots}$ can capture different

assumptions regarding the underlying seeding potential of the different types of cells that are present in the population. Here we consider three representative cases:

(a) **Seeding from the most advanced type**. When seeding happens before the introduction of the advanced type, the seeding cell is a type-0 cell, whereas when seeding happens after the introduction of type-1, the seeding cell is a type-1 cell. Concretely,

$$\phi_1^k = \begin{cases} 0, & k < T_1 \\ 1, & k \geq T_1 \end{cases}, \qquad \phi_0^k = 1 - \phi_1^k. \tag{13}$$

Such approach models the assumption that disseminated and circulating tumor cells are associated with advanced stage cancers and the rate of shedding generally increases with tumor size [48, 67].

(b) **Seeding proportional to fitness**. If $\lambda_0$ is the birth rate of type-0 cells and $\lambda_1$ the birth rate of type-1 cells,

$$\phi_1^k = \begin{cases} 0, & k < T_1 \\ \frac{\lambda_1}{\lambda_0 + \lambda_1}, & k \geq T_1 \end{cases}, \qquad \phi_0^k = 1 - \phi_1^k. \tag{14}$$

The coupling of dissemination and growth advantage is partly supported by the evidence that seeding cells more likely come from the tumor invasive front, where blood vessels are enriched [68].

(c) **Seeding uniformly over all living cells in the tumor**. In this approach the probability the seeding cell comes from a specific type is proportional to the number of cells in the population of such type. Concretely

$$\phi_1^k = \begin{cases} 0, & k < T_1 \\ \frac{N_1}{N_0 + N_1}, & k \geq T_1 \end{cases}, \qquad \phi_0^k = 1 - \phi_1^k, \tag{15}$$

where $N_0$, $N_1$ the population of cells of type 0 and 1, respectively, at seeding time. This scenario does not assume dissemination advantage of the new type, and the seeding is purely stochastic. This case is also considered here to account for the fact that few convincing recurrent genetic mutations have been identified so far that characteristically empower the metastatic dissemination in human cancers [11].

We can now focus on $B_{md}^k$. Consider first the case where the seeding cell originates from the type-0 cell population. When $k < T_1$ this event occurs with probability 1, whereas when $k \geq T_1$ it occurs with probability $\phi_0^k$. Notice that while different cells in the population can accumulate different passenger variants as the type-0 population grows over time, all cells share the same fitness characteristics, i.e., their death-to-birth ration is equal to $\rho_0$. Therefore, from a modeling perspective we fall within the setting analyzed in Section 4.1.2, and the expected $B_{md}^k$ can thus be given by (12) parametrized by $\rho = \rho_0$ and $f = \frac{2\alpha}{1-f_1}$. Similarly, if we condition on the seeding cell originating from type-1 we can calculate the expected $B_{md}$ based on (12) upon using the parametrization $\rho = \rho_1, f = \frac{2\alpha}{f_1}$ and $K = k_{\text{seed}} - T_1$. Therefore, a joint expression for

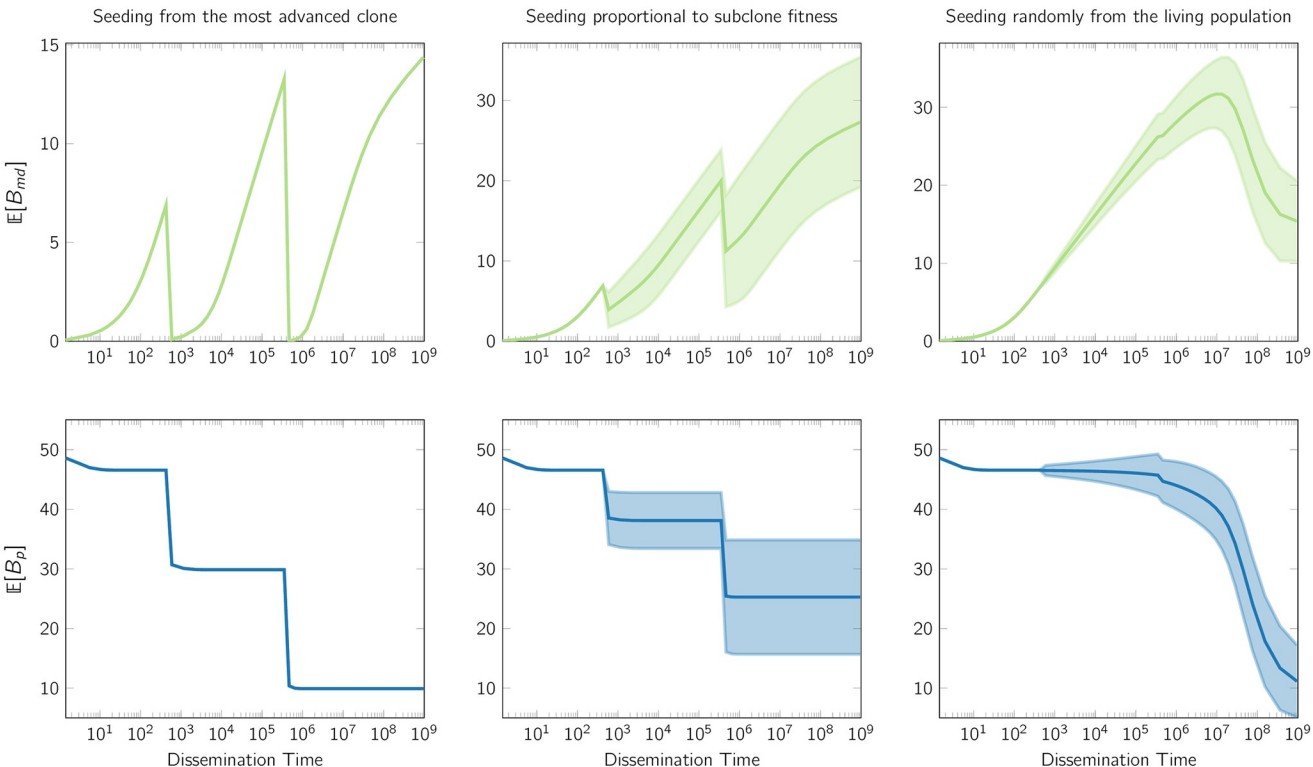

**Fig 7. Illustrative $B_{md}$ and $B_p$ patterns, in the exponential growth setting, under different seeding scenarios explored in Section 4.1.3.** In this example two advanced types of cells are introduced in the population; type-1 is introduced when there are 600 cells, whereas type-2 is introduced when there are approximately 460K cells. The parametric values are: $\rho_0 = 0.95$, $\rho_1 = 0.93$ $\rho_2 = 0.83$, $u = 0.15$, $f_m = 0.02$, $f_p = 0.4$; the overall population at detection time is $1.06 \times 10^9$ cells; finally, the advantageous types 1 and 2 are present on 87.3% and 77.1% of the cells in the population, respectively, and the number of clonal and subclonal variants of the primary tumor with frequency at least 40% is assumed to be 50. Type-2 is a subtype of Type-1. The variability reported on the subplots in the second and third column, captures the effects of uncertainty in seeding origination under seeding scenarios (b) and (c) (note that such variability is absent in the first column which corresponds to seeding scenario (a) which is deterministic; see also Section 4.1.3). Observe that the introduction of advanced cells results in a drop of the expected $B_{md}$ and $B_p$. This drop is more sudden and emphatic in the case where the probability of dissemination is strongly associated with the genetic properties of the seeding cells (as e.g., in the first two columns of the plot, which showcase the resulting patterns under seeding scenarios (a) and (b), discussed in Section 4.1.3), however, even on the other extreme (when seeding is equiprobable among all living cells at the time of dissemination; i.e., seeding scenario (c), detailed in Section 4.1.3), a gradual change in monotonicity of $B_{md}^k$ is also present. Similar patterns arise under different parametric choices (see S7 and S8 Figs).

approximating the expected $B_{md}$ in the case of two types is given by

$$\mathbb{E}[B_{md}; k, T_1, f_1] \approx$$
$$\approx \phi_1^k \times \mathbb{E}\left[B_{md}^{neutral}; k - T_1, \frac{2\alpha}{f_1}, \rho_1\right] + (1 - \phi_1^k) \times \mathbb{E}\left[B_{md}^{neutral}; k, \frac{2\alpha}{1 - f_1}, \rho_0\right]. \tag{16}$$

Note that the above approach can be extended to three or more types, in a straightforward way, and it thus omitted for the sake of brevity.

A fundamental implication of the above analysis is that the monotonicity of the sequence $\{\mathbb{E}[B_{md}; k]\}_{k=1,\dots}$ is expected to break by the introduction of newly emerging advantageous types. Such effect is, of course, inescapable when dissemination is assumed to be intertwined with fitness advantage (as in cases (a) and (b) described above), however the expected drop remains prominent even when no such assumption is enforced (Fig 7; top row, third column). Introduction of advantageous types affects $B_p$ in a way that mirrors the related effects on $B_{md}$. Notice that when new types lead to rapid drops in $B_{md}$, $B_p$ tracks these drops with abrupt decays (as seen in the bottom row of Fig 7), whereas when $B_{md}$ increases linearly with seeding

time, the corresponding $B_p$ regions appear flat. This is in accordance with our discussion in Section 4.1.1, and, in particular, with Eq (8). The variability due to the uncertainty in seeding origination, under scenarios (b) and (c), is increasing as new detectable types are introduced in the population—leading to the characteristic heteroscedastic patterns, also observed in Section 2.1. This is expected, as seeding from a new detectable type leads to a markedly smaller $B_{md}$, compared to seeding from previous types that are also present in the population. Note that this variability arises from the seeding origination distribution over time ($[\phi_0^k, \phi_1^k, \ldots]$) and is, of course, absent under seeding scenario (a) which is deterministic (Fig 7; first column). For additional plots reporting $B_{md}$ and $B_p$ patterns under different parametric choices, see S7 and S8 Figs.

## 4.2 Computational modeling

The primary goal of the above analysis was to reason mathematically about the expected behaviour of $B_{md}$ and $B_p$ across seeding times, as well as to derive example plots of such patterns for the case of analytically tractable growth models. Interestingly, such patterns remain true even under the more realistic setting, when tumor growth dynamics are governed by spatial 3D tumor structure (birth rate limitations imposed by local cell density etc). To study the behavior of the sought quantities at such more realistic settings we recur to our computational modeling framework, the details of which are presented below.

**4.2.1 Birth death process on a three-dimensional lattice.** We extend an existing single-cell-based spatial tumor growth model, named *Tumopp* [42], to simulate paired primary and metastatic tumors. Tumopp provides a flexible yet computationally efficient framework for investigating various parameter settings for tumor growth, making it an ideal foundation for our implementation. We name our simulation framework `Comet` (COmputational Modeling of Evolving Tumors). In our model, tumor expansion starts with a single founder cell located in the center of a three dimensional lattice with the Moore neighborhood. Cells divide at rate λ and die at rate $\mu$ per unit time. The waiting time between two consecutive division events is assumed to be exponentially distributed with mean 1/λ. We set λ = 0.25 by assuming that the average cell cycle is four days. The death rate is chosen according to experimentally estimated death-to-birth ratios $\rho = \mu/\lambda$ in human cancers. For the slow growing scenario (e.g., early primary tumor), we set $\mu = 0.2475$ to achieve a $\rho = 0.99$ [40], and for the fast growing scenario (e.g., metastatic tumor), we set $\mu = 0.18$ to achieve a $\rho = 0.72$ [39]. When a cell divides, one of its daughter cells stays at the original position on the grid, whereas the other fills one of the neighboring empty sites uniformly at random. When a cell dies, it is removed from the lattice. The local density is taken into consideration such that the actual birth rate of the $i$-th cell is $\lambda_i = \lambda \times \psi_i$, where $\psi_i$ is the proportion of empty neighboring sites for this cell. Intra-tumoral migration rate $m_i$ is set to be zero. This setting naturally reflects the spatial constraints in solid tumors and could lead to a pattern of peripheral growth consistent with experimental observations [43].

**4.2.2 Simulating various tumor expansion kinetics and metastatic seeding.** `Comet` allows passenger variants to occur at a rate of $u$ per cell birth. We randomly assign a virtual genomic coordinate for each variant by assuming that the variant resides in the broadly defined sequencing targetable region, such as the whole exome including UTRs ($5 \times 10^7$bp). Beneficial (or driver) variants which increase the birth rate by $s$, i.e., $\lambda_{daughter} = \lambda_{mother} \times (1 + s)$, occur at a rate $u_b$ per cell birth. The virtual genomic coordinate for a driver variant is sampled from a pool of $2 \times 10^5$bp in size. Although a smaller number of positions that confer fitness advantage have been assumed for point mutations [2], here we assume that a variant captures any type of somatic alterations, including the breakpoints of structural and copy number variants, as long as they can be resolved at the single nucleotide resolution. These settings are all

configurable and prepare our model to be comparable to other types of genomic divergence in the future. We employ various levels of the selection coefficient, in conjunction with the two levels of $\rho$, to model distinct growth dynamics. The selection coefficient is sampled from the Gaussian distribution with mean $s$ and standard deviation $s/2$. When $s = \{0, 0.02, 0.05\}$ and $\rho = 0.72$, the virtual tumor grows with a pattern of progressive diversification; $s = \{0.1, 0.2\}$ and $\rho = \{0.72, 0.99\}$ leads to a branching evolution and $s = \{0.5, 1\}$ and $\rho = 0.99$ realize the dynamics of linear evolution (Fig 5B). Although the parameter setting is model specific, the resulting clonal dynamics may also be realized by other forces that are not modeled here, such as micro-environmental forces [22] and purifying selection [69].

When the population size of the primary tumor increases by $N_s$ (25,000) cells ($N_d = I \times N_s$, where $I = 1, 2, \ldots$), we allow the next cell that is born, to disseminate and seed a metastatic tumor. This strategy takes into account both the impact of growth advantage and the existent population size of a subclone on metastatic dissemination potential. In other words, it falls between scenarios (b) seeding proportional to fitness and (c) seeding uniformly over all living cells in the tumor as discussed in the mathematical analysis in Section 4.1.3. Once a cell disseminates, it is removed from the lattice of the primary tumor, and is placed at the center of a new lattice devoted to the metastatic tumor. We also verified the model predictions by halving or doubling the value of $N_s$. We apply small $\rho$ (0.72) to metastatic tumor growth [39] and we let it inherit the rest of the parameter values of the corresponding primary tumor. We do not explore the dynamics of metastatic tumor expansion, as we focus on variants accrued in the seeding cell before dissemination, as well as variants in the primary tumor.

**4.2.3 Lineage tracing and virtual multi-region sampling.** Comet assigns a unique id to a cell when it is born, and tracks its location on the lattice, the id of its parent, the time of birth, the time when it divides or dies, as well as, the passenger and driver alterations (if any) occurred at cell's birth. Such detailed information allows us to reconstruct the full genealogy map and all the variants in every cell after simulation. Both the virtual primary and metastatic tumors reach a final volume of $N$ ($10^7$) cells. We use this number to strike a balance between computational efficiency in the single-cell-based setting, and practical relevance of the corresponding final physical size of the tumor [19]. Using this setting, we get a sequence of 400 virtual metastases as the primary tumor expands.

We randomly sample ten spherical regions (with a radius of 6.3 lattice units, leading to approximately 1,000 cells each) from the final primary and metastatic tumors, respectively. The variant repertoire of each region is reconstructed based on the genealogy relationship and variant history. A virtual sequencing experiment is performed to generate allele frequencies for each variant. The sequencing depth count is assumed to follow a negative binomial distribution [44] $D \sim \text{NegBinom}(m_d, \sigma_d)$, with mean $m_d = 100$, and dispersion $\sigma_d = 10$. The resulting read counts of the variant allele having a true allele frequency of $F$ follows a binomial distribution $M \sim \text{Binom}(D, F)$.

**4.2.4 Analysis of virtual genomic divergence and tumor genealogy.** The allele frequency of a variant $v$ in a region $r$ is defined as $f_v^r = M_v^r / D_v^r$. We evaluate the average between-region genomic divergence in the primary tumor based on the Fixation index (Fst) estimate [18] and the Fst between region $r$ and $r'$ is computed as follows:

$$\text{Fst}_{rr'} = \frac{\sum_{v=1}^{V} \left[ (f_v^r - f_v^{r'})^2 - \frac{f_v^r(1-f_v^r)}{D_v^r - 1} - \frac{f_v^{r'}(1-f_v^{r'})}{D_v^{r'} - 1} \right]}{\sum_{v=1}^{V} [f_v^r(1-f_v^{r'}) + f_v^{r'}(1-f_v^r)]}. \tag{17}$$

The genetic variance components (numerator and denominator) are averaged separately to obtain a ratio combining the Hudson Fst estimate [46] across all $V$ variants in the two regions. To obtain a global allele frequency of the variant $v$ in the entire tumor, we calculate a pooled

allele frequency across all the sampled regions for primary ($f_v^p = \sum M_v^r / \sum D_v^r$) and metastatic tumor ($f_v^m$), respectively [1]. The number of metastatic specific variants is calculated from the virtual sequencing data as $B_m = \#\{v|f_v^m > \gamma, f_v^p < \alpha\}$. Primary specific variants are calculated as $B_p = \#\{v|f_v^p > \gamma, f_v^m < \alpha\}$. In this study, we use substantial presence threshold $\gamma = 0.2$ and a sequencing detectability threshold $\alpha = 0.01$ to demonstrate our findings.

The clonal evolutionary tree of a virtual tumor is being analyzed as a directed graph with vertices denoting cells and edges representing birth events. The branch length between two cells (e.g., a seeding cell and its most recent detectable ancestor at frequency $\alpha$) is calculated as the number of edges between them. The total branch length of a group of cells (e.g., all detectable ancestors at frequency $\gamma$ in the primary tumor) is the sum of all edges in the sub-graph representing the genealogy of these cells.

To check if the subclonal expansion is associated with the drop of $B_m$, we first identified the time (e.g., the size of the primary tumor during its expansion) when a cell is born who eventually reach 5% of the final sampled population. To identify the time interval when $B_m$ drops, we then partitioned the zigzag pattern of running means of $B_m$ series to increasing and decreasing phases in each simulation by using the function ZigZag in R package `TTR` [70]. We also defined the valleys of $B_m$ as the time interval when $B_m$ lies below 5% of the means of the randomized data sampled (1000 times) from the original $B_m$ series. We then tested if the subclonal expansion is associated with the drop and valleys of $B_m$ significantly by performing a permutation test using the R package `regioneR` [71], by focusing on the interval where the primary tumor grows beyond 20% of its final size.

### 4.3 Code availability

Code for our single-cell based spatial tumor growth simulation studies and mathematical analysis is available at https://github.com/SunPathLab/Comet

## Supporting information

**S1 Appendix. Derivation of Formula (6) of the main document.**
(PDF)

**S1 Fig. Patterns of $B_p$.** (A) The single-cell evolutionary tree based definition of $B_p$: the total branch length of all Detectable Ancestors at $\gamma$ (DA$\gamma$) minus the branch length of the most recent detectable ancestor (MRDA$\gamma$) of the metastatic seeding cell, captures the majority of the variance of $B_p$ measured from the virtual tumors (blue density curves). By contrast, the tumor size at metastatic dissemination, as a surrogate of dissemination time (gray density curves) has much less explanatory value for $B_p$. Note that we focus on moderate and strong selection only, since the variability of $B_p$ under weak selection ($s < 0.05$) is negligible. (B) Two virtual tumors with distinct kinetics exemplify the conditional dependence of $B_p$ on metastatic dissemination time. The measured $B_p$ from virtual tumors is plotted against the tumor size fraction when the seeding cell (green dots) disseminates, the running mean (blue dots) and standard deviation (blue bars) of $B_p$ is also shown.
(TIF)

**S2 Fig. Heteroscedasticity of the time-ordered vector of M-P divergence.** The time-ordered vector of $B_m$ and $B_p$ are heteroscedastic under growth modes involving strong selection ($s \geq 0.1$). The p-value of White's test for heteroscedasticity [41] is shown at negative log10 scale. A p-value of 0.05 is marked as the dashed red vertical line.
(TIF)

**S3 Fig. The strong positive correlations between the mutation burden (or count) of the seeding cells and their Euclidean distance to the center of the 3D lattice.** We used the Euclidean distance of a cell to the center of the simulation lattice (Distance to Center) to plot the spatial cell genealogy trees (Figs 3B and 5B in the main manuscript). Here it is shown that the majority of the variance of this distance metric is captured by the mutation burden across our simulations (left panel). The right panel shows such strong positive correlations for two example simulation runs with distinct selection coefficients. This pattern is consistent with peripheral growth, attributable to the spatial constraints in the computational modeling. (TIF)

**S4 Fig. Pairwise variant allele frequency (VAF) histograms for virtual primary tumors.** Representative pairwise VAF histograms derived from three spatially separated regions (labeled Region 1 to 3) within the same virtual primary tumor (PT) are shown for tumors simulated under different evolutionary modes. The pairwise histograms illustrate the number of variants detected at a given VAF for the two tumor regions shown above and below the x axis. The bin width was optimized for visualization purposes on the basis of the number of variants [72]. Variants were classified as public (gray), shared (green, subclonal variants shared in the two regions in comparison), or region specific (blue) on the basis of their presence in the virtual multi-region sequencing (5 regions in total) [18]. The number of public variants, as well as the Fixation index (Fst) are indicated (see Methods). Whereas the between-region genetic divergence increases when we raise the selection coefficient from 0.02 to 0.05, raising the selection coefficient further facilitates late subclonal sweeps, which can decrease the between-region genetic divergence. Note the increasing number of public variants and decreasing Fst on the lower panel ($s = 0.2$ and $s = 0.5$). (TIF)

**S5 Fig. $B_{md}$ under neutral growth for different values of $f$.** (TIF)

**S6 Fig. $B_{md}$ variability for different values of $\rho$.** (TIF)

**S7 Fig. Example $B_{md}$ and $B_p$ plots for low death birth ratio.** The growth parameters of the example are: $\rho_0 = 0.72$; variant probability: $u = 0.15$; detectability frequency: $f = 0.02$; fitness of type 1: $\rho_1 = 0.64$; fitness of type 2: $\rho_2 = 0.12$. The population at detection time is $1.95e9$; type-1 mutation appears in 41.18% of the population; type-2 mutation appears in 20.49% of the population. (TIF)

**S8 Fig. Example $B_{md}$ and $B_p$ plots for high death birth ratio.** The growth parameters of the example are: $\rho_0 = 0.99$; variant probability: $u = 0.15$; detectability frequency: $f = 0.02$; fitness of type 1: $\rho_1 = 0.984$; fitness of type 2: $\rho_2 = 0.967$. The population at detection time is $1.30e9$; type-1 mutation appears in 91.84% of the population; type-2 mutation appears in 44.48% of the population. (TIF)

**S9 Fig. The mapping between seeding time and M-P divergence measured by running simulations with two alternative $N_s$.** The upper panel: $N_s = 12, 500$, i.e., a half of the original setting used in the main manuscript; the lower panel: $N_s = 50, 000$, i.e., a double of the original setting. We group the various simulations into three representative kinetics of primary tumor growth: progressive diversification ($s \leq 0.05$), branched evolution ($s = \{0.1, 0.2\}$) and linear evolution ($s \geq 0.5$). For each kinetics, the actual seeding time is shown as color scales on top of

the corresponding $B_m$ and $B_p$ values. A smoothed layer is added to show the general distribution of the seeding time.
(TIF)

**S10 Fig. The mapping between seeding time and M-P divergence measured by sampling five random regions of the virtual tumors.** We group the various simulations into three representative kinetics of primary tumor growth: progressive diversification ($s \leq 0.05$), branched evolution ($s = \{0.1, 0.2\}$) and linear evolution ($s \geq 0.5$). For each kinetics, the actual seeding time is shown as color scales on top of the corresponding $B_m$ and $B_p$ values. A smoothed layer is added to show the general distribution of the seeding time.
(TIF)

## Acknowledgments

This study uses the computing resources in Minnesota Supercomputing Institute. We thank Dr. Jasmine Foo and Zicheng Wang for commenting on the analytical framework. We also thank Dr. Boyang Liu and Dr. Jie Ding for helpful discussions.

## Author Contributions

**Conceptualization:** Ruping Sun, Athanasios N. Nikolakopoulos.

**Data curation:** Ruping Sun, Athanasios N. Nikolakopoulos.

**Formal analysis:** Ruping Sun, Athanasios N. Nikolakopoulos.

**Funding acquisition:** Ruping Sun.

**Investigation:** Ruping Sun, Athanasios N. Nikolakopoulos.

**Methodology:** Ruping Sun, Athanasios N. Nikolakopoulos.

**Project administration:** Ruping Sun.

**Resources:** Ruping Sun.

**Software:** Ruping Sun, Athanasios N. Nikolakopoulos.

**Supervision:** Ruping Sun.

**Validation:** Ruping Sun, Athanasios N. Nikolakopoulos.

**Visualization:** Ruping Sun, Athanasios N. Nikolakopoulos.

**Writing – original draft:** Ruping Sun, Athanasios N. Nikolakopoulos.

**Writing – review & editing:** Ruping Sun, Athanasios N. Nikolakopoulos.

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
