## [Decision Letter · Decision Letter 0]

4 Dec 2020

Dear Dr. Sun,

Thank you very much for submitting your manuscript "Elements and Evolutionary Determinants of Genomic Divergence Between Paired Primary and Metastatic Tumors" for consideration at PLOS Computational Biology.

As with all papers reviewed by the journal, your manuscript was reviewed by members of the editorial board and by several independent reviewers. In light of the reviews (below this email), we would like to invite the resubmission of a significantly-revised version that takes into account the reviewers' comments.

The reviewers have all appreciated the value of your work but they collectively have a range of suggestions to improve the manuscript. The most substantial criticism is perhaps that of Reviewer 1 who asks you to make a more direct comparison between simulation and analytical results. Reviewers 2 and 3 make several suggestions for improving the presentation of all aspects of the manuscript from the background to exposition of theory to results. It should be possible for you to address all of these comments in a revision. I agree with Reviewer 2 in particular that the manuscript is hard to follow in many places. Please make your revised manuscript more accessible to people in and outside your area of specialisation.

We cannot make any decision about publication until we have seen the revised manuscript and your response to the reviewers' comments. Your revised manuscript is also likely to be sent to reviewers for further evaluation.

Sincerely,

Mark M. Tanaka

Associate Editor

PLOS Computational Biology

Natalia Komarova

Deputy Editor

PLOS Computational Biology

The reviewers have all appreciated the value of your work but they collectively have a range of suggestions to improve the manuscript. The most substantial criticism is perhaps that of Reviewer 1 who asks you to make a more direct comparison between simulation and analytical results. Reviewers 2 and 3 make several suggestions for improving the presentation of all aspects of the manuscript from the background to exposition of theory to results. It should be possible for you to address all of these comments in a revision. I agree with Reviewer 2 in particular that the manuscript is hard to follow in many places. Please make your revised manuscript more accessible to people in and outside your area of specialisation.

Reviewer's Responses to Questions

**Comments to the Authors:**

Reviewer #1: Elements and Evolutionary Determinants of Genetic Divergence Between Paired Primary and Metastatic Tumors

By Ruping Sun and Athanasios N. Nikolakopoulos

The authors investigated how the observed genetic divergence between paired primary and metastatic tumors is regulated. I believe the purpose of this paper is scientifically and medically meaningful to understand the genetic relationship between primary and metastatic tumors from paired genomic data that infers the time of seeding, which might lead to a proposal of the best treatment option at diagnosis in the future. They performed mathematical analysis and computational simulations separately and revealed that dissemination from an advanced detectable subclone of the primary tumor can lead to an abrupt drop in the expected measurable M-P divergence, thereby breaking the previously assumed monotonic relation between seeding time and M-P divergence. I am not convinced with their major finding stated above because they showed two kinds of results derived from mathematical analysis and computational simulations separately and then they insisted these are consistent without comparing them in the same panel. I would like to see clear correspondence between mathematical predictions and computational simulations with the same parameter values in the same panel. After that, the analytical insights become convincing to support the conclusion.

Moreover, their conclusion that a non-monotonic relationship appears under branched and linear evolution does not have a big impact since it is just a denial, rather I expect the authors propose a rule of the genetic divergence between primary and metastatic tumor under such evolution.

Reviewer #2: please see attachment

Reviewer #3: Overall this is a nicely written paper that needs some minor modifications before being acceptable for publication.

First off, I'm surprised that the authors do not expound more about multi-region sequencing in the last paragraph of the introduction. Surely the use of this approach that is common in the literature of studying evolutionary dynamics in primary tumors is worth mentioning its benefits and limitations as well. Especially since this is discussed briefly in the discussion (lines 305-307 and again in lines 318-320). I argue that multi-region sequencing has begun to ensure that spatial sampling locations are preserved, albeit with poor resolution, but exists, nonetheless.

Line 329-333 the authors use poor resolution sample heterogeneity estimates as justification for their lack of comparisons between real data and computational models. While this is certainly true. The authors should include a sentence to propose how this could be resolved in future studies to begin to think about how data can be used to validate the authors primary findings and justifications.

The authors state, based on citation 30, that the majority of distant mets are monoclonal. The citation of a single article is insufficient given several other studies that have shown different cancer types that result in various mets showing evidence for polyclonal seeding (e.g. Hirotsu 2020, Hu 2020). At minimum this should be discussed in greater detail. However, I feel it’s necessary to have polyclonal seedings combined with the monoclonal seedings the authors describe. If the “analytical thinking is extendable” (Line 364 to 366), this should be done to provide greater insights into our understandings of metastasis.

Figure 5A needs some adjustments to be interpretable. First and foremost move the selection coefficient outside of the plotted area or put a distinctive box around this. The points appear as part of the plot. The points in this legend should be aligned by the center of the circle as opposed to the bottom. For six discreet groupings of variance a more divergent colormap would aid in visualization. The intervals themselves within this plot are odd; why are they different? Jumping from 0.1 intervals to 0.05 intervals and back to 0.1 intervals?

Section 4.2.4. There seems a slight disconnect with the authors use of virtual sequencing versus the Hudson Fst estimate and the multi-region sampling above in 4.2.3. This isn’t virtual sequencing in a way that can relate back to any real world data with technical error, sequencing depth, and attempts to being able to relate any of the metrics/calculations back to real data. This section would be better described only by ‘Tumor genealogy analysis.’ Virtual sequencing would be more akin to methods proposed by Williams et al. 2018 and several other papers. The descriptions in 4.2.3 where assumed negative binomial distribution with depth being accounted for are present here. What differs in these two sections in their samplings and assumptions for calculating VAFs?

Minor Comments

Remove the comma in the sentence starting on line 40 (A recently published, deme…).

Add a comma after however in line 298.

**Have all data underlying the figures and results presented in the manuscript been provided?**

Reviewer #1: Yes

Reviewer #2: **No: **as we ask in our comments, the code does not seem to be available

Reviewer #3: None

PLOS authors have the option to publish the peer review history of their article (what does this mean?). If published, this will include your full peer review and any attached files.

Reviewer #1: No

Reviewer #2: **Yes: **Jacob G Scott

Reviewer #3: **Yes: **Alexander R. A. Anderson
---

## [Decision Letter · Decision Letter 1]

26 Feb 2021

Dear Dr. Sun,

We are pleased to inform you that your manuscript 'Elements and Evolutionary Determinants of Genomic Divergence Between Paired Primary and Metastatic Tumors' has been provisionally accepted for publication in PLOS Computational Biology.

Best regards,

Mark M. Tanaka

Associate Editor

PLOS Computational Biology

Natalia Komarova

Deputy Editor

PLOS Computational Biology

The reviewers are all satisfied with the revised manuscript. Thank you for the effort you put into addressing all their comments.

Reviewer's Responses to Questions

**Comments to the Authors:**

Reviewer #1: I am satisfied with the authors' responses to my comments. I think the paper is ready for the publication.

Reviewer #2: Thank you for your excellent revision.

Reviewer #3: The authors have satisfied my concerns and bolstered their discussions with how their work impacts and relates to the field at large. Specifically, the authors have expanded their discussion points on polyclonal seeding of metastasis, multi-region sequencing, and clarifications provided for in-silico sequencing, plots, and text.

**Have all data underlying the figures and results presented in the manuscript been provided?**

Reviewer #1: Yes

Reviewer #2: Yes

Reviewer #3: Yes

PLOS authors have the option to publish the peer review history of their article (what does this mean?). If published, this will include your full peer review and any attached files.

Reviewer #1: No

Reviewer #2: **Yes: **Jacob G Scott

Reviewer #3: **Yes: **Alexander R. A. Anderson

---

## [Editor Report · Acceptance letter]

13 Mar 2021

PCOMPBIOL-D-20-01821R1 

Elements and Evolutionary Determinants of Genomic Divergence Between Paired Primary and Metastatic Tumors

Dear Dr Sun,

I am pleased to inform you that your manuscript has been formally accepted for publication in PLOS Computational Biology. Your manuscript is now with our production department and you will be notified of the publication date in due course.

With kind regards,

Alice Ellingham
